# Transformers Generalize DeepSets and Can be Extended to Graphs and Hypergraphs

**Jinwoo Kim, Saeyoon Oh, Seunghoon Hong**
School of Computing, KAIST
{jinwoo-kim, saeyoon17, seunghoon.hong}@kaist.ac.kr

## Abstract

We present a generalization of Transformers to any-order permutation invariant data (sets, graphs, and hypergraphs). We begin by observing that Transformers generalize DeepSets, or first-order (set-input) permutation invariant MLPs. Then, based on recently characterized higher-order invariant MLPs, we extend the concept of self-attention to higher orders and propose higher-order Transformers for order-$k$ data ($k = 2$ for graphs and $k > 2$ for hypergraphs). Unfortunately, higher-order Transformers turn out to have prohibitive complexity $\mathcal{O}(n^{2k})$ to the number of input nodes $n$. To address this problem, we present sparse higher-order Transformers that have quadratic complexity to the number of input hyperedges, and further adopt the kernel attention approach to reduce the complexity to linear. In particular, we show that the sparse second-order Transformers with kernel attention are theoretically more expressive than message passing operations while having an asymptotically identical complexity. Our models achieve significant performance improvement over invariant MLPs and message-passing graph neural networks in large-scale graph regression and set-to-(hyper)graph prediction tasks. Our implementation is available at `https://github.com/jw9730/hot`.

## 1 Introduction

Graph is a universal data modality used to model social networks [28], chemical compounds [10], biological structures [9], and interactions in particle physics [21, 29]. Recent graph neural networks (GNNs) adopt a message-passing scheme [39, 37, 32], where the node features are propagated and aggregated recurrently according to the neighborhood structure given in the input graph. Despite the simplicity, the local and recurrent nature makes them unable to discover dependency between any two nodes with a distance longer than the message-passing steps [11]. The locality of message-passing is also known to be related to the over-smoothing problem that prevents scaling of GNNs [23, 4, 27].

An alternative approach is using a more general set of operations that involve global interactions while respecting the permutation symmetry of graphs. Such operations either produce the same output regardless of the node ordering of input graph (permutation invariance), or commute with node reordering (permutation equivariance). Message-passing is an equivariant operation, but is restricted to local neighborhoods defined by the input graph. Recently, Maron et. al. (2019) [26] characterized the full space of invariant and equivariant linear layers. It turned out that these layers span not only the local neighborhood interactions of message-passing GNNs, but also global interactions such as the one between disconnected nodes, and even edge-to-node, or node-to-edge interactions (Figure 1). Notably, their formulation naturally extends to layers with different input & output orders (e.g., edge in, node out), and higher-order layers for hypergraphs. In theory, an invariant MLP composed of these layers should be more powerful than message-passing GNNs as they can model long-range dependency between nodes (Figure 1). However, they are currently not widely adopted due to relatively low performance and high asymptotic memory complexity.

35th Conference on Neural Information Processing Systems (NeurIPS 2021).

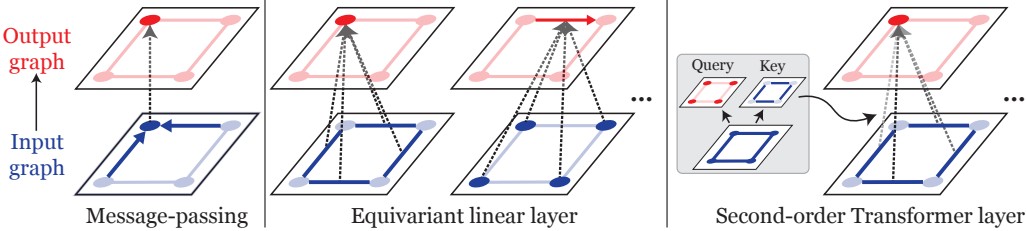

Figure 1: Illustrated operations of a message-passing GNN, an equivariant linear layer, and a second-order Transformer layer. A single output node is highlighted.

**Contributions**  In this work, we present higher-order Transformers that address the low performance and high complexity of invariant MLPs. First, based on an observation that the renowned Transformer encoder generalizes first-order equivariant linear layers or DeepSets [35], we formulate higher-order Transformer layers that generalize equivariant linear layers by extending self-attention to higher orders (Figure 1). Second, while Transformer layers with input and output orders $k, l$ has $\mathcal{O}(n^{k+l})$ asymptotic complexity, we show that by leveraging the sparsity of input hypergraphs, we can obtain $\mathcal{O}(m^2)$ complexity given input with $m$ hyperedges. Further adopting kernel attention approaches, we propose a variant that reduces the complexity to $\mathcal{O}(m)$ and theoretically prove that they are more expressive than message passing networks. Finally, we test higher-order Transformers on a range of tasks, and demonstrate that they achieve significant improvements over invariant MLPs and are highly competitive in performance and scalability to the state-of-the-art GNNs.

## 2   Preliminary

In this section, we first describe how (hyper)graphs can be treated as higher-order tensors. We then describe linear layers that operate on higher-order tensors while respecting node permutation symmetry. In particular, we analyze operations within linear layers for second-order tensors (graphs) and show they involve global interactions on top of local aggregation.

Let us define some notations. We denote a set as $\{a, ..., b\}$, a tuple as $(a, ..., b)$, and denote $[n] = \{1, ..., n\}$. We denote the space of order-$k$ tensors as $\mathbb{R}^{n^k \times d}$ where $d$ is feature dimension. For an order-$k$ tensor $\mathbf{A} \in \mathbb{R}^{n^k \times d}$, we use multi-index $\mathbf{i} = (i_1, ..., i_k) \in [n]^k$ to denote $\mathbf{A_i} = \mathbf{A}_{i_1,...,i_k} \in \mathbb{R}^d$. Let $S_n$ be the set of all permutations of $[n]$. $\pi \in S_n$ acts on $\mathbf{i}$ by $\pi(\mathbf{i}) = (\pi(i_1), ..., \pi(i_k))$, and acts on an order-$k$ tensor $\mathbf{A}$ by $(\pi \cdot \mathbf{A})_\mathbf{i} = \mathbf{A}_{\pi^{-1}(\mathbf{i})}$.

**(Hyper)graphs as tensors**  Generally, a (hyper)graph data $G$ can be represented as a tuple $(V, \mathbf{A})$, where $V$ is a set of $n$ nodes and $\mathbf{A} \in \mathbb{R}^{n^k \times d}$ encodes features attached to hyperedges. The type of the hypergraph is indicated by the order $k$ of the tensor $\mathbf{A}$. First-order tensor is a set of features (e.g., point cloud, bag-of-words) where $\mathbf{A}_i$ is the feature of node $i$. Second-order tensor encodes edge features (e.g., adjacency) where $\mathbf{A}_{i_1,i_2}$ is the feature of edge $(i_1, i_2)$. Generally, an order-$k$ tensor encodes hyperedge features (e.g., mesh) where $\mathbf{A}_{i_1,...,i_k}$ is the feature of hyperedge $(i_1, ..., i_k)$.

**Permutation invariance and equivariance**  Our problem of interest is building a functional relation $f(\mathbf{A}) \approx T$ between tensor $\mathbf{A}$ and target $T$. If $T$ is a single output vector, we often require that $f$ is *permutation invariant*, that it satisfies $f(\pi \cdot \mathbf{A}) = f(\mathbf{A})$; if $T$ is a tensor $T = \mathbf{T}$, we often require that $f$ is *permutation equivariant*, satisfying $f(\pi \cdot \mathbf{A}) = \pi \cdot f(\mathbf{A})$, for all $\pi \in S_n$ and $\mathbf{A} \in \mathbb{R}^{n^k \times d}$. In a typical design setup where a neural network $f$ is built using linear layers and non-linear activations, the construction of $f$ reduces to finding invariant and equivariant *linear* layers.

**Invariant and equivariant linear layers**  We summarize invariant linear layers $L_{k \to 0} : \mathbb{R}^{n^k \times d} \to \mathbb{R}^{d'}$ and equivariant linear layers $L_{k \to l} : \mathbb{R}^{n^k \times d} \to \mathbb{R}^{n^l \times d'}$ identified in Maron et. al. (2019) [26]. Note that invariant layer is a special case of $L_{k \to l}$ with $l = 0$. In summary, given an input $\mathbf{A} \in \mathbb{R}^{n^k \times d}$, the order-$l$ output of an equivariant layer $L_{k \to l}$ can be written with $\mathbf{i} \in [n]^k, \mathbf{j} \in [n]^l$ as:

$$L_{k \to l}(\mathbf{A})_\mathbf{j} = \sum_\mu \sum_\mathbf{i} \mathbf{B}^\mu_{\mathbf{i},\mathbf{j}} \mathbf{A}_\mathbf{i} w_\mu + \sum_\lambda \mathbf{C}^\lambda_\mathbf{j} b_\lambda, \tag{1}$$

where $w_\mu \in \mathbb{R}^{d \times d'}$, $b_\lambda \in \mathbb{R}^{d'}$ are weight and bias parameters, $\mathbf{B}^\mu \in \mathbb{R}^{n^{k+l}}$ and $\mathbf{C}^\lambda \in \mathbb{R}^{n^l}$ are *basis tensors* (will be defined), and $\mu$ and $\lambda$ are *equivalence classes* of order-$(k + l)$ and order-$l$

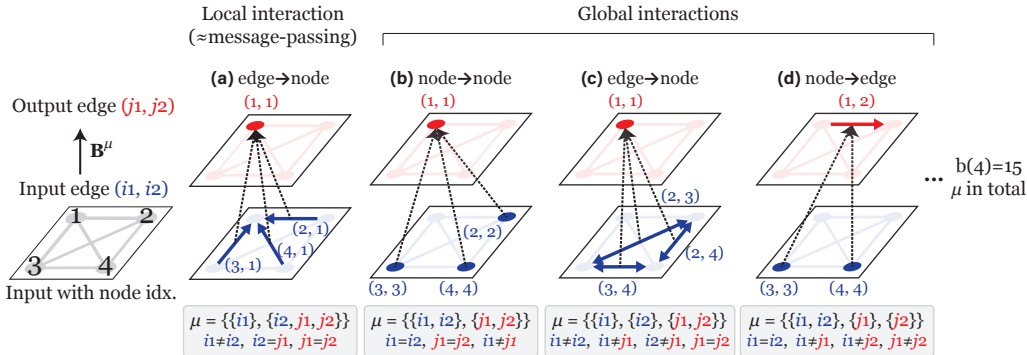

Figure 2: Example operations in a second-order layer $L_{2 \to 2}$. We illustrate how input edges $(i_1, i_2) = \mathbf{i}$ are aggregated to an output edge $(j_1, j_2) = \mathbf{j}$ in various equivalence classes $(\mathbf{i}, \mathbf{j}) \in \mu$ via basis tensor $\mathbf{B}_{\mathbf{i},\mathbf{j}}^{\mu}$. Note that a loop $((i_1, i_1)$ or $(j_1, j_1))$ represents a node.

multi-indices, respectively. The equivalence classes are defined upon equivalence relation $\sim$ that, for multi-indices $\mathbf{i}, \mathbf{j} \in [n]^k$, $\mathbf{i} \sim \mathbf{j}$ iff $(i_1, ..., i_k) = (\pi(j_1), ..., \pi(j_k))$ for some node permutation $\pi \in S_n$. Then, a multi-index $\mathbf{i}$ and all members $\mathbf{j}$ in its equivalence class have the same (permutation-invariant) equality pattern: $\mathbf{i}_a = \mathbf{i}_b \Leftrightarrow \mathbf{j}_a = \mathbf{j}_b$ for all $a, b \in [k]$. Consequently, each equivalence class $\mu$ (or $\lambda$) is a distinct set of all order-$(k + l)$ (or order-$l$) multi-indices having a specific equality pattern.

Notably, we can represent each equivalence class of order-$k$ multi-indices as a unique partition of $[k]$ regardless of $n$, where the partition specifies the equality pattern. *e.g.*, with $k = 2$, we have two partitions and respective equivalent classes: $\mu_1 = \{\{i_1, i_2\}\}$ the set of all $(i_1, i_2)$ with $i_1 = i_2$, and $\mu_2 = \{\{i_1\}, \{i_2\}\}$ the set of all $(i_1, i_2)$ with $i_1 \neq i_2$. Thus, with b$(k)$ the $k$-th Bell number or number of partitions of $[k]$, we have b$(k+l)$ equivalence classes $\mu$ for the weight and b$(l)$ equivalence classes $\lambda$ for the bias. We guide the readers interested in derivation to Maron et. al. (2019) [26].

With the equivalence classes, basis tensors are then defined as follows:

$$\mathbf{B}_{\mathbf{i},\mathbf{j}}^{\mu} = \left\{ \begin{array}{ll} 1 & (\mathbf{i}, \mathbf{j}) \in \mu \\ 0 & \text{otherwise} \end{array} \right. \quad ; \quad \mathbf{C}_{\mathbf{j}}^{\lambda} = \left\{ \begin{array}{ll} 1 & \mathbf{j} \in \lambda \\ 0 & \text{otherwise} \end{array} \right. \quad (2)$$

In Eq. (1), each equivalence class $\mu$ determines which (hyper)edges participate and how they interact in summation $\sum_{\mathbf{i}} \mathbf{B}_{\mathbf{i},\mathbf{j}}^{\mu} \mathbf{A}_{\mathbf{i}}$. As an example, let us consider $L_{2 \to 2}$ that maps input edges $\mathbf{i} = (i_1, i_2)$ to output edges $\mathbf{j} = (j_1, j_2)$. An equivalence class $\mu = \{\{i_1, i_2\}, \{j_1, j_2\}\}$ represents all $(\mathbf{i}, \mathbf{j}) \in \mu$ that $i_1 = i_2$, $j_1 = j_2$, and $i_1 \neq j_1$. Then, due to masking by $\mathbf{B}^{\mu}$, only input elements $\mathbf{A}_{i_1, i_2}$ with $i_1 = i_2$ participate in the summation and gives output $L_{2 \to 2}(\mathbf{A})_{j_1, j_2}$ for $j_1 = j_2$, $i_1 \neq j_1$. Intuitively, this is analogous to computing a node feature by sum-pooling all the other nodes (Fig. 2(b)). Different $\mu$ accounts for other interactions as shown in Figure 2. Notably, it contains a richer set of operations beyond local interactions modeled by message-passing (Fig. 2(a)), such as global interaction across all nodes (Fig. 2(b)), and edge-to-node (Fig. 2(c)), node-to-edge interactions (Fig. 2(d)).

We finish the section by writing out the first-order equivariant layer $L_{1 \to 1}$. As b$(2) = 2$, the layer has two equivalence classes $\mu_1 = \{\{i_1, j_1\}\}$ and $\mu_2 = \{\{i_1\}, \{j_1\}\}$. Then, we have $\mathbf{B}^{\mu_1} = I_n$ and $\mathbf{B}^{\mu_2} = 1_n 1_n^{\top} - I_n$, with $1_n \in \mathbb{R}^n$ vector of ones. Then, given a set of features $\mathbf{A} \in \mathbb{R}^{n \times d}$,

$$L_{1 \to 1}(\mathbf{A}) = I_n \mathbf{A} w_1' + (1_n 1_n^{\top} - I_n) \mathbf{A} w_2' + 1_n b^{\top} \quad (3)$$

$$= I_n \mathbf{A} w_1 + 1_n 1_n^{\top} \mathbf{A} w_2 + 1_n b^{\top}, \quad (4)$$

where $w_1, w_2, w_1', w_2' \in \mathbb{R}^{d \times d'}$, $b \in \mathbb{R}^{d'}$. $L_{1 \to 1}$ is analogous to a combination of elementwise feedforward ($\mu_1$) and sum-pooling of set elements ($\mu_2$), and is also known as a DeepSet layer [35].

## 3 Higher-Order Transformers

In Section 2, we introduced higher-order linear equivariant layers $L_{k \to l}$, and showed that they contain various global and node/edge interactions that are not covered by message-passing. In this section, we establish a connection between the first-order equivariant layer $L_{1 \to 1}$ and self-attention of Transformer encoder layers [31]. Then, we extend the relationship to higher orders by tensorizing queries and keys, and formulate higher-order Transformer layers. We finish the section by proposing a principled parameter reduction for queries and keys, which reduces a fair amount of computation.

## 3.1 Transformers generalize DeepSets

As shown in Section 2, first-order linear layer, or DeepSet, has a simple structure composed of feedforward and sum-pooling (Eq. (4)). Although it is theoretically proven to be a universal approximator of permutation-invariant functions [35], static sum-pooling could be limited in capturing interactions of set elements, motivating the use of sophisticated pooling. In particular, the self-attention mechanism of Transformer encoder [31] was shown to achieve a large performance gain in set modeling via context-aware weighted pooling [22, 34]. To see this, let us write out the Transformer encoder layers.

A Transformer encoder layer is a function $\text{Enc} : \mathbb{R}^{n \times d} \to \mathbb{R}^{n \times d}$ consisting of two layers: a self-attention layer $\text{Attn} : \mathbb{R}^{n \times d} \to \mathbb{R}^{n \times d}$ and an elementwise feedforward layer $\text{MLP} : \mathbb{R}^{n \times d} \to \mathbb{R}^{n \times d}$. For a set of $n$ input vectors $X \in \mathbb{R}^{n \times d}$, a Transformer layer computes the following:

$$\text{Attn}(X)_i = X_i + \sum_{h=1}^{H} \sum_{j=1}^{n} \alpha_{ij}^h X_j w_h^V w_h^O, \tag{5}$$

$$\text{Enc}(X)_i = \text{Attn}(X)_i + \text{MLP}(\text{Attn}(X))_i, \tag{6}$$

where $\alpha^h = \sigma(X w_h^Q (X w_h^K)^\top)$ is an attention coefficient with an activation $\sigma$, $H$ is the number of heads, $d_H$ is head size, $d_F$ is hidden dimension, and $w_h^O \in \mathbb{R}^{d_H \times d}$, $w_h^V, w_h^K, w_h^Q \in \mathbb{R}^{d \times d_H}$.[1]

Now, we show that Transformer layers are generalized first-order linear equivariant functions. By setting $\alpha_{ij}^h = 1$ and assuming that $\text{MLP}(Y)$ approximates a linear layer $Y w^F + b^{F\top}$ following the universal approximation theorem [16], the Transformer layer reduces to the following [2] :

$$\text{Enc}(X)_i = X_i(I_n + w^F) + \sum_{j=1}^{n} X_j w^{VO}(I_n + w^F) + b^{F\top}, \tag{7}$$

where $w^{VO} = \sum_{h=1}^{H} w_h^V w_h^O$. This is equivalent to a DeepSet layer in Eq. (4) with $w_1 = I_n + w^F$, $w_2 = w^{VO}(I_n + w^F)$, and $b = b^F$. In other words, we can convert DeepSets to Transformers by changing the static pooling to attention and replacing elementwise linear mapping with an MLP. We generalize this approach to higher-order input and output, which leads to the formulation of higher-order Transformers for graphs and hypergraphs.

## 3.2 Higher-order Transformer layers

In Section 3.1, we showed that Transformer layers are generalized first-order equivariant linear layers $L_{1 \to 1}$. Notably, the generalization procedure was equivalent to changing static pooling to attention and adding feedforward MLP. In this section, we generalize the approach to $L_{k \to l}$ with arbitrary orders $k$ and $l$ and formulate higher-order Transformer layer $\text{Enc}_{k \to l}$.

In general, we define higher-order Transformer layer as a function $\text{Enc}_{k \to l} : \mathbb{R}^{n^k \times d} \to \mathbb{R}^{n^l \times d}$ with two layers: a higher-order self-attention layer $\text{Attn}_{k \to l} : \mathbb{R}^{n^k \times d} \to \mathbb{R}^{n^l \times d}$ and a feedforward layer $\text{MLP}_{l \to l} : \mathbb{R}^{n^l \times d} \to \mathbb{R}^{n^l \times d}$. For an input tensor $\mathbf{A} \in \mathbb{R}^{n^k \times d}$, a Transformer layer computes:

$$\text{MLP}_{l \to l}(\text{Attn}_{k \to l}(\mathbf{A})) = L_{l \to l}^2 \left( \text{ReLU}(L_{l \to l}^1(\text{Attn}_{k \to l}(\mathbf{A}))) \right), \tag{8}$$

$$\text{Enc}_{k \to l}(\mathbf{A}) = \text{Attn}_{k \to l}(\mathbf{A}) + \text{MLP}_{l \to l}(\text{Attn}_{k \to l}(\mathbf{A})), \tag{9}$$

where $L_{l \to l}^1 : \mathbb{R}^{n^l \times d \to n^l \times d_F}$ and $L_{l \to l}^2 : \mathbb{R}^{n^l \times d_F \to n^l \times d}$ are equivariant linear layers with hidden dimension $d_F$. Remaining question is how to define and compute higher-order self-attention $\text{Attn}_{k \to l}(\mathbf{A})$.

To design $\text{Attn}_{k \to l}$, we remove the bias from Eq. (1) and introduce attention coefficients. It is achieved by changing each $\mathbf{B}^\mu \in \mathbb{R}^{n^{k+l}}$ to an attention coefficient tensor $\boldsymbol{\alpha}^{h,\mu} \in \mathbb{R}^{n^{k+l}}$ with multiple heads:

$$\text{Attn}_{k \to l}(\mathbf{A})_{\mathbf{j}} = \sum_{h=1}^{H} \sum_{\mu} \sum_{\mathbf{i}} \boldsymbol{\alpha}_{\mathbf{i},\mathbf{j}}^{h,\mu} \mathbf{A_i} w_{h,\mu}^V w_{h,\mu}^O, \tag{10}$$

where $w_{h,\mu}^O \in \mathbb{R}^{d_H \times d}$, $w_{h,\mu}^V \in \mathbb{R}^{d \times d_H}$ are learnable parameters, $H$ denotes the number of heads, and $d_H$ denotes head size. Then similar to first-order case (Section 3.1), we can show the following:

---

[1]Note that we omitted normalization after $\text{Attn}(\cdot)$ and $\text{MLP}(\cdot)$ for simplicity as in [34, 12].

[2]In practice, Transformer employs softmax in attention and deviates from Deepsets due to normalization.

**Theorem 1.** *$Enc_{k \to l}$ (Eq. (9)) is a generalization of $L_{k \to l}$ (Eq. (1)).*

*Proof.* Let $\boldsymbol{\alpha}_{\mathbf{i,j}}^{h,\mu} = 1$ for all $h, \mu$, and $(\mathbf{i,j}) \in \mu$. This renders $\boldsymbol{\alpha}^{h,\mu} = \mathbf{B}^\mu$ from the definition of $\mathbf{B}^\mu$. Additionally, let $\mathrm{MLP}_{l \to l}(\mathrm{Attn}_{k \to l}(\mathbf{A}))_{\mathbf{j}} = \sum_\lambda \mathbf{C}_{\mathbf{j}}^\lambda b_\lambda$ for all $\mathbf{j} \in [n]^l$. That is, $\mathrm{MLP}_{l \to l}$ ignores input and reduces to an invariant bias in Eq. (1). Then, Eq. (9) reduces to the following:

$$\mathrm{Enc}_{k \to l}(\mathbf{A})_{\mathbf{j}} = \sum_\mu \sum_{\mathbf{i}} \mathbf{B}_{\mathbf{i,j}}^\mu \mathbf{A}_{\mathbf{i}} \sum_{h=1}^H w_{h,\mu}^V w_{h,\mu}^O + \sum_\lambda \mathbf{C}_{\mathbf{j}}^\lambda b_\lambda, \tag{11}$$

which is equivalent to Eq. (1) with $w_\mu = \sum_{h=1}^H w_{h,\mu}^V w_{h,\mu}^O$. $\qquad\square$

Now, we describe how to compute each attention tensor $\boldsymbol{\alpha}^\mu \in \mathbb{R}^{n^{k+l}}$ from input $\mathbf{A} \in \mathbb{R}^{n^k \times d}$ (Eq. (10), we drop head index $h$ for brevity). We obtain each attention tensor from higher-order query and key:

$$\boldsymbol{\alpha}_{\mathbf{i,j}}^\mu = \begin{cases} \sigma(\mathbf{Q}_{\mathbf{j}}^\mu, \mathbf{K}_{\mathbf{i}}^\mu)/Z_{\mathbf{j}} & (\mathbf{i,j}) \in \mu \\ 0 & \text{otherwise} \end{cases} \text{ where } \mathbf{Q}^\mu = L_{k \to l}^\mu(\mathbf{A}), \mathbf{K}^\mu = L_{k \to k}^\mu(\mathbf{A}). \tag{12}$$

where $Z_{\mathbf{j}} = \sum_{\mathbf{i}|(\mathbf{i,j}) \in \mu} \sigma(\mathbf{Q}_{\mathbf{j}}^\mu, \mathbf{K}_{\mathbf{i}}^\mu)$ is a normalization constant. Note that query and key tensors are computed from the input $\mathbf{A}$ using the equivariant linear layers in Eq. (1). This leads to permutation equivariance (or invariance) of Transformer encoder layer $\mathrm{Enc}_{k \to l}$ in Eq. (9).

**Reducing orders of query and key** Although Eq. (10) and Eq. (12) provide a simple and generic definition of higher-order self-attention, we observe that there exist a lot of unnecessary computations. Specifically, there exist elements of query $\mathbf{Q}^\mu$ and key $\mathbf{K}^\mu$ that are unused in computation of attention coefficient $\boldsymbol{\alpha}_{\mathbf{i,j}}^\mu$, as it depends only on indices satisfying $(\mathbf{i,j}) \in \mu$. In fact, it turns out that the effective orders of query and key are much smaller than $l$ and $k$, as we show below:

**Proposition 1.** *From Eq. (12), let $u(\cdot)$ denote the number of unique entries in a multi-index. With $u_q = u(\mathbf{j})$, $u_k = u(\mathbf{i})$ for some $(\mathbf{i,j}) \in \mu$, we can always find suitable linear layers $L_{k \to u_q}^\mu$, $L_{k \to u_k}^\mu$ and index space mappings $f_q^\mu : [n]^l \to [n]^{u_q}$, $f_k^\mu : [n]^k \to [n]^{u_k}$ that satisfy the following.*

$$\boldsymbol{\alpha}_{\mathbf{i,j}}^\mu = \sigma(\tilde{\mathbf{Q}}_{\mathbf{j}'}^\mu, \tilde{\mathbf{K}}_{\mathbf{i}'}^\mu)/\tilde{Z}_{\mathbf{j}} \; \forall (\mathbf{i,j}) \in \mu, \tag{13}$$

*where $\tilde{Z}_{\mathbf{j}} = \sum_{\mathbf{i}|(\mathbf{i,j}) \in \mu} \sigma(\tilde{\mathbf{Q}}_{\mathbf{j}'}^\mu, \tilde{\mathbf{K}}_{\mathbf{i}'}^\mu)$, $\tilde{\mathbf{Q}}^\mu = L_{k \to u_q}^\mu(\mathbf{A})$, $\tilde{\mathbf{K}}^\mu = L_{k \to u_k}^\mu(\mathbf{A})$, $\mathbf{j}' = f_q^\mu(\mathbf{j})$, $\mathbf{i}' = f_k^\mu(\mathbf{i})$.*

*Proof.* We leave the proof in Appendix A.1.1. $\qquad\square$

Based on Proposition 1, we can compute query and key in Eq. (12) in a much compact way using linear layers with output orders $u_q$ and $u_k$ instead of $l$ and $k$. In our experiments, we observe that this optimization is very useful in reducing the number of parameters and memory footprint to a feasible level without affecting the effective model capacity.

## 4 Asymptotically Efficient Higher-Order Transformers

In Section 3, we formulated higher-order Transformer layers $\mathrm{Enc}_{k \to l}$ and showed that they generalize linear equivariant layers $L_{k \to l}$. However, this capability comes with a cost; the high asymptotic complexity of the Transformer encoder limits its practical merits. Specifically, we show the following:

**Property 1.** *Given input size $n$, the asymptotic complexity of a linear layer $L_{k \to l}$ (Eq. (1)) is $\mathcal{O}(n^{k+l})$, and complexity of an encoder layer $\mathrm{Enc}_{k \to l}$ (Eq. (9)) is $\mathcal{O}(n^{k+l} + n^{2k} + n^{2l})$.*

*Proof.* We leave the proof in Appendix A.1.2. $\qquad\square$

Thus, in this section, we further analyze the encoder layer and propose a number of optimization and relaxation to reduce the asymptotic complexity with a minimal impact on capability. Notably, combining all our strategies reduces the complexity to $\mathcal{O}(m)$ given a hypergraph with $m$ hyperedges. Even with such efficiency, we show that the reduced version of higher-order Transformer is theoretically more expressive than all message-passing neural networks.

## 4.1 Linear layers with reduced complexity

A major computation bottleneck within Transformer encoder layer $\text{Enc}_{k \to l}$ is the higher-order linear layer, since it is used for key, query, and feedforward layer. By exploiting only a subset of basis, we show that we can implement lightweight version of linear layer with reduced asymptotic complexity.

**Proposition 2.** *Given a linear layer $L_{k \to l}$ in Eq. (1), we can always find a nonempty subset $\mathcal{M}$ of equivalence classes such that computation of the following for all $\mathbf{j}$ has $\mathcal{O}(n^l)$ complexity.*

$$\bar{L}_{k \to l}(\mathbf{A})_{\mathbf{j}} = \sum_{\mu \in \mathcal{M}} \sum_{\mathbf{i}} \mathbf{B}_{\mathbf{i},\mathbf{j}}^{\mu} \mathbf{A}_{\mathbf{i}} w_\mu + \sum_{\lambda} \mathbf{C}_{\mathbf{j}}^{\lambda} b_\lambda, \tag{14}$$

*Proof.* We leave the proof in Appendix A.1.3. $\qquad\square$

We term the reduced linear layer $\bar{L}_{k \to l}$ in Eq. (14) as a *lightweight* linear layer. In practice, we choose $\mathcal{M}$ among equivalence classes that do not involve summation over input. For instance, for $\bar{L}_{1 \to 1}$, we use only the basis for elementwise mapping ($I_n$) and drop sum-pooling ($1_n 1_n^\top$) from Eq. (4). This approximation effectively reduces the complexity of linear layers at the cost of losing inter-element dependencies. We employ the lightweight linear layers within the $\text{Enc}_{k \to l}$ (Eq. (9)) to compute query and key embeddings, while the element dependency within $\text{Enc}_{k \to l}$ is handled by the higher-order self-attention using all equivalence classes as in Eq. (12). This design is coherent to original (first-order) Transformers, where the elements are first linearly projected to query/key with elementwise basis ($I_n$) and interaction is handled by attention (implicitly $1_n 1_n^\top$). Importantly, this does not hurt Theorem 1 as attention coefficients can still reduce to one and MLP can reduce to bias.

By using lightweight linear layers in $\text{Enc}_{k \to l}$, we can significantly reduce the computational cost. The complexity of $L_{k \to u_k}^\mu$, $L_{k \to u_q}^\mu$, and $\text{MLP}_{l \to l}$ in Eq. (13) and Eq. (9) reduces to $\mathcal{O}(n^k)$, $\mathcal{O}(n^l)$, and $\mathcal{O}(n^l)$, respectively, and as a result $\text{Enc}_{k \to l}$ becomes $\mathcal{O}(n^{k+l})$. As a result, we have higher-order Transformer layers $\text{Enc}_{k \to l}$ that generalize $L_{k \to l}$ while retaining the complexity $\mathcal{O}(n^{k+l})$. From here we assume that all linear layers for key, query, and MLP are lightweight.

## 4.2 Sparse Transformer layers

Even with lightweight linear layers, $\mathcal{O}(n^{k+l})$ complexity of $\text{Enc}_{k \to l}$ is still impractical. For example, for graphs, complexity larger than $\mathcal{O}(n^2)$ is regarded prohibitive while $\text{Enc}_{2 \to 2}$ is $\mathcal{O}(n^4)$. Fortunately, leveraging the sparsity inherent in real-world graphs can significantly reduce the complexity. As we show, it reduces the complexity of $\text{Enc}_{k \to l}$ to $\mathcal{O}(m^2)$ for processing a hypergraph with $m$ hyperedges.

Let $E$ the set of hyperedges of input hypergraph $G$. Each hyperedge is generally represented by a multi-index $\mathbf{i} \in [n]^k$, so we denote $E = \{\mathbf{i}_1, ..., \mathbf{i}_m\}$ with $m$ hyperedges. Leveraging sparsity of $E$ is straightforward when the order of the hypergraph is fixed (e.g., $L_{k \to k}$); we can perform computations with only respect to the existing hyperedges $\mathbf{i}, \mathbf{j} \in E$. However, in our framework, order can change by layers (e.g., $L_{k \to l}$), making it difficult to directly transfer the sparsity structure $E$ to different-order output. Inspired by network projection [5], we remedy this by constructing $E'$ such that for any $\mathbf{j} \in E'$, there exists $\mathbf{i} \in E$ containing all unique elements of $\mathbf{j}$. For example, for $L_{3 \to 2}$, this corresponds to projection of third-order $E$ to second-order $E'$ by obtaining edges from sides of triangles. Despite simplicity, this simple heuristic works well in general and generalizes to any order.

Then, we integrate the hyperedge sets $E, E'$ into computation of linear layer in Eq. (14) as follows:

$$\bar{L}_{k \to l}(\mathbf{A}, E)_{\mathbf{j}} = \begin{cases} \sum_{\mu \in \mathcal{M}} \sum_{\mathbf{i} \in E} \mathbf{B}_{\mathbf{i},\mathbf{j}}^{\mu} \mathbf{A}_{\mathbf{i}} w_\mu + \sum_{\lambda} \mathbf{C}_{\mathbf{j}}^{\lambda} b_\lambda & \mathbf{j} \in E' \\ 0 & \text{otherwise} \end{cases} \tag{15}$$

Likewise, integrating $E, E'$ into attention computation in Eq. (10) and Eq. (12) we have:

$$\text{Attn}_{k \to l}(\mathbf{A}, E)_{\mathbf{j}} = \begin{cases} \sum_{h=1}^{H} \sum_{\mu} \sum_{\mathbf{i} \in E} \boldsymbol{\alpha}_{\mathbf{i},\mathbf{j}}^{h,\mu} \mathbf{A}_{\mathbf{i}} w_{h,\mu}^V w_{h,\mu}^O & \mathbf{j} \in E' \\ 0 & \text{otherwise} \end{cases} \tag{16}$$

$$\text{where } \boldsymbol{\alpha}_{\mathbf{i},\mathbf{j}}^{h,\mu} = \begin{cases} \sigma(\mathbf{Q}_{\mathbf{j}}^{h,\mu}, \mathbf{K}_{\mathbf{i}}^{h,\mu})/Z_{\mathbf{j}} & (\mathbf{i}, \mathbf{j}) \in \mu, \mathbf{i} \in E, \mathbf{j} \in E' \\ 0 & \text{otherwise} \end{cases}, \tag{17}$$

where $\mathbf{Q}^{h,\mu} = \bar{L}_{k \to l}^{h,\mu}(\mathbf{A}, E)$, $\mathbf{K}^{h,\mu} = \bar{L}_{k \to k}^{h,\mu}(\mathbf{A}, E)$, $Z_{\mathbf{j}} = \sum_{\mathbf{i} | (\mathbf{i},\mathbf{j}) \in \mu \wedge \mathbf{i} \in E} \sigma(\mathbf{Q}_{\mathbf{j}}^{h,\mu}, \mathbf{K}_{\mathbf{i}}^{h,\mu})$.

With the computations, we can show the following:

**Property 2.** *When given $E$ with $m$ elements, the equivariant linear layer in Eq. (15) has $\mathcal{O}(m)$ complexity, and self-attention computation in Eq. (16) has $\mathcal{O}(m^2)$ complexity. Consequently, the computation done by a $\text{Enc}_{k \to l}$ composed of the layers has $\mathcal{O}(m^2)$ complexity.*

*Proof.* We leave the proof in Appendix A.1.5. $\qquad\square$

## 4.3 Kernel attention trick

Section 4.2 shows that, by constraining linear layers within $Enc_{k\to l}$ to have sparse input and output, we can reduce $\mathcal{O}(n^{k+l})$ complexity to quadratic $\mathcal{O}(m^2)$ to input size. Yet, even $\mathcal{O}(m^2)$ can be demanding with large or dense input. As the quadratic term comes from self-attention computation, we follow the prior work in kernel attention [18, 6] and view attention coefficients as pairwise dot-product scores. As we will show, this allows us to further reduce the complexity of $Enc_{k\to l}$ to $\mathcal{O}(n^k + n^l)$ and even $\mathcal{O}(m)$ for sparse version, at the cost of relaxing some modeling assumption.

We begin by approximating attention coefficient in Eq. (12) using pairwise dot-product kernel [18, 6]:

$$\boldsymbol{\alpha}_{\mathbf{i},\mathbf{j}}^{\mu} = \left\{ \begin{array}{ll} \phi(\mathbf{Q}_{\mathbf{j}}^{\mu})^{\top}\phi(\mathbf{K}_{\mathbf{i}}^{\mu})/Z_{\mathbf{j}} & (\mathbf{i},\mathbf{j}) \in \mu \\ 0 & \text{otherwise} \end{array} \right. \quad \text{where } Z_{\mathbf{j}} = \sum_{\mathbf{i}|(\mathbf{i},\mathbf{j})\in\mu} \phi(\mathbf{Q}_{\mathbf{j}}^{\mu})^{\top}\phi(\mathbf{K}_{\mathbf{i}}^{\mu}) \quad (18)$$

where $\phi : \mathbb{R}^{d_H} \to \mathbb{R}_{+}^{d_K}$ is kernel feature map. The choice of kernel can be flexible, and in our implementation we adopt Performer kernel [6] that has strong theoretical and empirical guarantee.

Substituting Eq. (18) in Eq. (10), we have:

$$\text{Attn}_{k\to l}(\mathbf{A})_{\mathbf{j}} = \sum_{\mu} Z_{\mathbf{j}}^{-1} \sum_{\mathbf{i}|(\mathbf{i},\mathbf{j})\in\mu} \phi(\mathbf{Q}_{\mathbf{j}}^{\mu})^{\top}\phi(\mathbf{K}_{\mathbf{i}}^{\mu})\mathbf{A}_{\mathbf{i}}w_{\mu}^{V}w_{\mu}^{O}$$

$$= \sum_{\mu} Z_{\mathbf{j}}^{-1}\phi(\mathbf{Q}_{\mathbf{j}}^{\mu})^{\top} \sum_{\mathbf{i}|(\mathbf{i},\mathbf{j})\in\mu} \phi(\mathbf{K}_{\mathbf{i}}^{\mu})\mathbf{A}_{\mathbf{i}}w_{\mu}^{V}w_{\mu}^{O}, \quad (19)$$

$$\text{where } Z_{\mathbf{j}} = \phi(\mathbf{Q}_{\mathbf{j}}^{\mu})^{\top} \sum_{\mathbf{i}|(\mathbf{i},\mathbf{j})\in\mu} \phi(\mathbf{K}_{\mathbf{i}}^{\mu}). \quad (20)$$

Here, the inner-summations in Eq. (19) and Eq. (20) are over the key index $\mathbf{i}$, which is *coupled* with the query index $\mathbf{j}$ by $(\mathbf{i},\mathbf{j}) \in \mu$. This coupling causes the major computational bottleneck, since the inner-summations should be computed for every $\mathbf{j}$-th query. We propose an approximation by decoupling the key and query indices and taking inner-summations over $\mathcal{I} = \bigcup_{\mathbf{j}} \{\mathbf{i}|(\mathbf{i},\mathbf{j}) \in \mu\}$:

$$\text{Attn}_{k\to l}(\mathbf{A})_{\mathbf{j}} \approx \sum_{\mu} Z_{\mathbf{j}}^{-1}\phi(\mathbf{Q}_{\mathbf{j}}^{\mu})^{\top} \sum_{\mathbf{i}\in\mathcal{I}} \phi(\mathbf{K}_{\mathbf{i}}^{\mu})\mathbf{A}_{\mathbf{i}}w_{\mu}^{V}w_{\mu}^{O}, \text{ where } Z_{\mathbf{j}} \approx \phi(\mathbf{Q}_{\mathbf{j}}^{\mu})^{\top} \sum_{\mathbf{i}\in\mathcal{I}} \phi(\mathbf{K}_{\mathbf{i}}^{\mu}). \quad (21)$$

The approximation allows a query to attend to some additional keys (depending on $\mu$), but it does not hurt the central requirement of attention that each query can assign different attention weights to the keys. With the approximation, we can compute the summations $\sum_{\mathbf{i}\in\mathcal{I}} \phi(\mathbf{K}_{\mathbf{i}}^{\mu})\mathbf{A}_{\mathbf{i}}$ and $\sum_{\mathbf{i}\in\mathcal{I}} \phi(\mathbf{K}_{\mathbf{i}}^{\mu})$ only once and reuse them across all query indices $\mathbf{j}$ to reduce the cost. Specifically, we show:

**Property 3.** *The encoder $Enc_{k\to l}$ with approximation in Eq. (21) has a complexity of $\mathcal{O}(n^k + n^l)$. Exploiting sparsity further reduces the complexity to $\mathcal{O}(m)$, linear to the number of hyperedges $m$.*

*Proof.* We leave the proof in Appendix A.1.6. ☐

## 4.4 Theoretical analysis and comparison to message-passing

We showed that exploiting sparsity and adopting kernel attention reduces the computational complexity of the Transformer encoder to linear to input edges. Considering graphs ($k = l = 2$), this complexity is equivalent to or better than $\mathcal{O}(n + m)$ complexity of the message passing operation[3]. Still, we show that our (approximate) encoder is theoretically more expressive than message passing.

Specifically, we show the following:

**Theorem 2.** *A composition of two sparse Transformer layers $Enc_{2\to 2}$ with kernel attention can approximate any message passing algorithms (Gilmer et. al. (2017) [10]) to arbitrary precision, while the opposite is not true.*

*Proof.* We leave the proof in Appendix A.1.7. ☐

This leads to the following corollary:

**Corollary 1.** *Second-order sparse Transformers with kernel attention are more expressive than any message-passing neural networks within the framework of Gilmer et. al. (2017) [10].*

In the proof, we note that local information propagation in message-passing GNNs is carried out in $Enc_{2\to 2}$ by a single $\mu = \{\{i_1\}, \{i_2, i_3, i_4\}\}$. Other types of $\mu$ would carry out different operations, which provides intuition on the powerfulness of Transformers.

---

[3]In practice, we place node features on the diagonals of the adjacency matrix, leading to $\mathcal{O}(n + m)$.

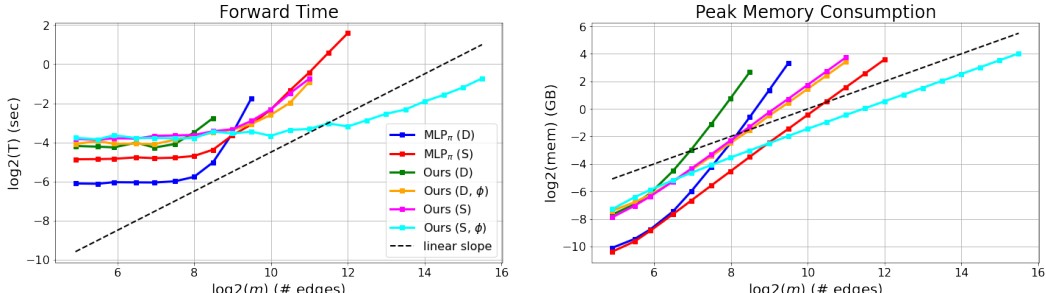

Figure 3: Comparison of all second-order models in terms of forward time, memory consumption, and maximal possible input size. Plots are shown until each model runs into out-of-memory error on a RTX 6000 GPU with 22GB.

## 5  Experiments

In this section, we demonstrate the capability of higher-order Transformers on a variety of tasks including synthetic data, large-scale graph regression, and set-to-(hyper)graph prediction. Specifically, we use a synthetic node classification dataset from Gu et. al. (2020) [11], a molecular graph regression dataset from Hu et. al. (2021) [17], two set-to-graph prediction datasets from Serviansky et. al. (2020) [29], and three hyperedge prediction datasets used in Zhang et. al. (2020) [36]. Details including the datasets and hyperparameters can be found in Appendix A.2. We implemented invariant MLP in Maron et. al. (2019) [26] as one of baselines, which we abbreviate as $MLP_\pi$. We build our model by gradually adding lightweight linear layers (Sec. 4.1), sparse linear layers (Sec. 4.2), and kernel attention (Sec. 4.3) and denote them by (D), (S), and ($\phi$), respectively.

**Runtime and memory analysis**   To experimentally verify our claims on linear complexity in Sec. 4.3, we conducted a runtime and memory consumption analysis on all second-order models using random graphs; details can be found in Appendix A.2.3. The results are shown in Figure 3. Consistent with our theoretical claims, sparse second-order Transformer with kernel attention (Ours (S, $\phi$)) is the only variant that linearly scales to input size in terms of both time and memory. Also, it is the only one that successfully scales to graphs with $> 50k$ edges, while still very fast in smaller graphs.

**Synthetic chains**   To test the ability of higher-order Transformers in modeling long-range interactions in graphs, we used a synthetic dataset where the task is node classification in chain graphs. Binary class information is provided in a terminal node, so a model is required to propagate the information across the chain by handling long-range dependency. We used training and test sets with chains of length 20 and 200 respectively. As baselines, we used 3 message-passing networks, a sparse invariant MLP, and ablated versions of sparse second-order Transformers where $\mu$ for global pooling are removed (w/o global). Further details can be found in Appendix A.2.4.

The test performances are in Table 1. We first note that second-order Transformers successfully capture long-range dependency up to 200 nodes apart, while message-passing networks fail. Importantly, when the subset of basis that accounts for global pooling is eliminated, the performance of Transformers drops similar to message-passing networks, showing their importance in modeling long-range interaction. Yet, simply having global basis is not enough, as seen in the failure of $MLP_\pi$. This indicates fine-grained interaction modeling via attention is essential even in this simple task.

Table 1: Chain node classification results.

| Method | Micro-$F_1$ (%) | Macro-$F_1$ (%) |
|---|---|---|
| GCN | $47.78 \pm 4.17$ | $33.58 \pm 1.86$ |
| GIN-0 | $53.72 \pm 4.17$ | $36.22 \pm 1.86$ |
| GAT | $47.78 \pm 4.17$ | $33.58 \pm 1.86$ |
| $MLP_\pi$ (S) | $53.5 \pm 4.16$ | $36.04 \pm 1.97$ |
| Ours (S) w/o global | $53.72 \pm 4.17$ | $36.22 \pm 1.86$ |
| Ours (S, $\phi$) w/o global | $50.77 \pm 5.15$ | $35.22 \pm 2.17$ |
| Ours (S) | $\mathbf{100 \pm 0}$ | $\mathbf{100 \pm 0}$ |
| Ours (S, $\phi$) | $\mathbf{100 \pm 0}$ | $\mathbf{100 \pm 0}$ |

**Large-scale graph regression**   To further evaluate higher-order Transformers in large-scale setting, we used the PCQM4M-LSC dataset from Open Graph Benchmark [17], which is the largest graph-level regression dataset composed of 3.8M molecular graphs. As test data is unavailable, we report the Mean Absolute Error (MAE) on validation dataset. In addition to the baselines from the benchmark, we also report the performances of second-order invariant MLP and a vanilla (first-order) Transformer[4] for comparison. Further details can be found in Appendix A.2.5.

---

[4]As vanilla Transformer operates on node features only, we used Laplacian graph embeddings [2, 8] as positional embeddings so that the model can consider edge structure information.

The results are in Table 2. Second-order Transformer outperforms the message-passing GNNs (GCN, GIN) by a large margin, including the ones with a virtual node that can model long-range interactions (GCN-VN, GIN-VN). It suggests that higher-order attention is potentially better in handling long-range interactions on graphs than the current practice of augmenting GNNs with a virtual node. Furthermore, second-order Transformer outperforms invariant MLP, indicating that replacing sum-pooling with attention is important for scale-up. Finally, second-order Transformer significantly outperforms vanilla Transformer with Laplacian graph embeddings. This is presumably because node embeddings are insufficient to utilize features associated with edges, while second-order Transformers can naturally use all edge information. We also note that invariant MLP and vanilla Transformer have worse complexity than the second-order Transformer ($\mathcal{O}(m^2)$ and $\mathcal{O}(n^2)$, respectively), while the second-order Transformer has $\mathcal{O}(m)$ complexity identical to GCN.

Table 2: PCQM4M-LSC large-scale graph regression results. * indicates results are obtained with a shorter schedule (10% of the full iterations).

| Model | Validate MAE |
|---|---|
| MLP-FINGERPRINT ([17]) | 0.2044 |
| GCN ([17]) | 0.1684 |
| GIN ([17]) | 0.1536 |
| GCN-VN ([17]) | 0.1510 |
| GIN-VN ([17]) | 0.1396 |
| Transformer + Laplacian PE* | 0.2162 |
| $MLP_\pi$ (S)* | 0.1464 |
| Ours (S, $\phi$)$_{-SMALL}$* | 0.1376 |
| Ours (S, $\phi$)* | 0.1294 |
| Ours (S, $\phi$) | **0.1263** |

**Set-to-graph prediction** An important advantage of our framework distinguished from most existing GNNs is that, by design, it can be applied to any learning scenario with different input and output orders (mixed-order). To demonstrate this, we tested higher-order Transformers in set-to-graph prediction tasks where the goal is to predict edge structure of a graph given a set of node features. We used two datasets following the prior work [29]. The first dataset Jets originates from particle physics experiments, where collision of high-energy particles gives a set of observed particles. The task is to partition the feature set of observed particles according to their origin. By viewing each subset of particles as a fully-connected graph, the problem is cast as a set-to-graph prediction. For the second dataset Delaunay, the task is to predict Delaunay triangulation [7] given a set of points in 2D space. Two datasets are used for this task, one containing 50 points and the other containing varying number of points $\in \{20, ..., 80\}$. For the baselines, we take the scores reported in Serviansky et. al. (2020) [29], which includes GNNs and invariant MLPs (S2G, S2G+) with $L_{1\rightarrow1}$ and $L_{1\rightarrow2}$. Our model is made by substituting the linear layers with $Enc_{1\rightarrow1}$ and $Enc_{1\rightarrow2}$ (mixed-order) respectively. Further details including the datasets, metrics, and baselines can be found in Appendix A.2.6.

The results are outlined in Table 3. Mixed-order Transformers, both softmax and kernel attention, have favorable scores over all baselines. Especially, they outperform all baselines by a large margin in Delaunay: note that GNNs fall into trivial solution with high recall but very low precision. We particularly note that the Transformers' performance in Delaunay (20-80) is comparable to Delaunay (50), with $0.3$-$0.7\%$ drop in F1 score. Compared with S2G that exhibits $\sim 12\%$ drop, this indicates attention mechanism within $Enc_{1\rightarrow2}$ is helpful in modeling varying number of nodes.

Table 3: Set-to-graph results. Lower-right panel shows Delaunay (20-80) sample from ours and S2G.

| | Method | F1 | RI | ARI | | Method | Acc | Prec | Rec | F1 |
|---|---|---|---|---|---|---|---|---|---|---|
| Jets (B) | AVR | 0.565 | 0.612 | 0.318 | Delaunay (50) | SIAM | 0.939 | 0.766 | 0.653 | 0.704 |
| | MLP | 0.533 | 0.643 | 0.315 | | SIAM-3 | 0.911 | 0.608 | 0.538 | 0.570 |
| | SIAM | 0.606 | 0.675 | 0.411 | | GNN0 | 0.826 | 0.384 | 0.966 | 0.549 |
| | SIAM-3 | 0.597 | 0.673 | 0.396 | | GNN5 | 0.809 | 0.363 | **0.985** | 0.530 |
| | GNN | 0.586 | 0.661 | 0.381 | | GNN10 | 0.759 | 0.311 | 0.978 | 0.471 |
| | S2G | 0.646 | 0.736 | 0.491 | | S2G | 0.984 | 0.927 | 0.926 | 0.926 |
| | S2G+ | 0.655 | 0.747 | 0.508 | | S2G+ | 0.983 | 0.927 | 0.925 | 0.926 |
| | Ours (D) | 0.667 | 0.746 | 0.520 | | Ours (D) | **0.994** | **0.981** | 0.967 | **0.974** |
| | Ours (D, $\phi$) | **0.670** | **0.751** | **0.526** | | Ours (D, $\phi$) | 0.991 | 0.967 | 0.952 | 0.959 |
| Jets (C) | AVR | 0.695 | 0.650 | 0.326 | Delaunay (20-80) | SIAM | 0.919 | 0.667 | 0.764 | 0.687 |
| | MLP | 0.686 | 0.658 | 0.319 | | SIAM-3 | 0.895 | 0.578 | 0.622 | 0.587 |
| | SIAM | 0.729 | 0.695 | 0.406 | | GNN0 | 0.810 | 0.387 | 0.946 | 0.536 |
| | SIAM-3 | 0.719 | 0.710 | 0.421 | | GNN5 | 0.777 | 0.352 | **0.975** | 0.506 |
| | GNN | 0.720 | 0.689 | 0.390 | | GNN10 | 0.746 | 0.322 | 0.970 | 0.474 |
| | S2G | 0.747 | 0.727 | 0.457 | | S2G | 0.947 | 0.736 | 0.934 | 0.799 |
| | S2G+ | 0.751 | 0.733 | 0.467 | | S2G+ | 0.947 | 0.735 | 0.934 | 0.798 |
| | Ours (D) | 0.755 | 0.732 | 0.469 | | Ours (D) | **0.993** | **0.982** | 0.960 | **0.971** |
| | Ours (D, $\phi$) | **0.757** | **0.735** | **0.473** | | Ours (D, $\phi$) | 0.989 | 0.948 | 0.956 | 0.952 |
| Jets (L) | AVR | 0.970 | 0.965 | 0.922 | | | | | | |
| | MLP | 0.960 | 0.957 | 0.894 | | | | | | |
| | SIAM | 0.973 | 0.970 | 0.925 | | | | | | |
| | SIAM-3 | 0.895 | 0.876 | 0.729 | | | | | | |
| | GNN | 0.972 | 0.970 | 0.929 | | | | | | |
| | S2G | 0.972 | 0.970 | 0.931 | | | | | | |
| | S2G+ | 0.971 | 0.969 | 0.929 | | | | | | |
| | Ours (D) | **0.974** | **0.972** | **0.935** | | | | | | |
| | Ours (D, $\phi$) | **0.974** | **0.972** | **0.935** | | | | | | |

Ground Truth    Ours (D,φ)    S2G    — FP   — FN

Table 4: $k$-uniform hyperedge prediction results. For Hyper-SAGNN, we reproduced the scores using the open-sourced code. For additional baselines including node2vec, we take the scores reported in Zhang et. al. (2020) [36].

| | GPS | | MovieLens | | Drug | |
|---|---|---|---|---|---|---|
| | AUC | AUPR | AUC | AUPR | AUC | AUPR |
| node2vec-mean ([36]) | 0.563 | 0.191 | 0.562 | 0.197 | 0.670 | 0.246 |
| node2vec-min ([36]) | 0.570 | 0.185 | 0.539 | 0.186 | 0.684 | 0.258 |
| DHNE ([36]) | 0.910 | 0.668 | 0.877 | 0.668 | 0.925 | 0.859 |
| Hyper-SAGNN-E | 0.947 | 0.788 | 0.922 | **0.792** | 0.963 | 0.897 |
| Hyper-SAGNN-W | 0.907 | 0.632 | 0.909 | 0.683 | 0.956 | 0.890 |
| S2G+ (S) | 0.943 | 0.726 | 0.918 | 0.737 | 0.963 | 0.898 |
| Ours (S, $\phi$) | **0.952** | **0.804** | **0.923** | 0.771 | **0.964** | **0.901** |

$k$**-uniform hyperedge prediction**    One major advantage of our framework is that it naturally extends to higher-order data (hypergraphs). To demonstrate this, we consider higher-order extension of set-to-graph prediction task, where the goal is predicting $k$-uniform hyperedges (e.g., user-location-activity) from node features. For evaluation, we used three datasets for transductive 3-edge prediction following the prior work [36]. The first dataset GPS derives from a GPS network [38], and contains (user-location-activity) hyperedges. The second dataset MovieLens is a social network dataset of tagging activities [13], containing (user-movie-tag) hyperedges. The third dataset Drug comes from a medicine network from FAERS[5], containing (user-drug-reaction) hyperedges. As baselines, we consider higher-order invariant MLP (S2G+) and the state-of-the-art, self-attention based Hyper-SAGNN [36]. We implemented higher-order Transformer and S2G+ by substituting $\text{Enc}_{1\to2}$ and $L_{1\to2}$ in set-to-graph architectures to $\text{Enc}_{1\to3}$ and $L_{1\to3}$, respectively. Further details including the datasets, metrics, and baselines can be found in Appendix A.2.2 and Appendix A.2.7.

The results are in Table 4. Higher-order Transformer outperforms S2G+ in all datasets, and Hyper-SAGNN in all but one metric in MovieLens. The results suggest that higher-order self-attention is effective in learning higher-order representation beyond second-order graphs. The results are encouraging especially because we did not introduce any form of task-specific heuristics into the model, while some of the baselines such as Hyper-SAGNN depend on many inductive biases (static/dynamic branches, Hadamard power, etc.).

## 6    Discussion

In this paper, we proposed a generalization of Transformers to higher-orders, and applied a number of design strategies that reduce their complexity to a feasible level. Higher-order Transformers are attractive, both in theory and application. In theoretical aspect, it inherits the theoretical completeness and expressive power of invariant MLPs. In application aspect, it is potentially more powerful than message-passing neural networks due to global interaction modeling, and can be extended to a variety of useful mixed-order tasks involving sets, graphs, and hypergraphs.

At the same time, our work has some limitations that need to be addressed in future work. First, although complexity to input size can be lowered to linear, the number of basis grows rapidly with increasing order due to $\mathcal{O}((0.792k/\ln(k+1))^k)$ asymptotic formula of $k$-th Bell number [3], still making the model infeasible in higher orders. Improvement approaches such as finding a compact subset of basis that retains universality [29] and exploiting unorderedness [26] are promising in this direction. Second, our work builds upon tensor-based representation of graphs, which makes it difficult to be directly extended to hypergraphs containing edges with varying orders (e.g., co-citation networks). This is a common challenge to all tensor-based methods [14, 26, 25, 19, 29], and we believe addressing this would be an important future research direction.

**Acknowledgements**    This work was supported in part by National Research Foundation of Korea (NRF) grant funded by the Korea government (MSIT) (2021R1C1C1012540, 2021R1A4A3032834) and Institute of Information & Communications Technology Planning & Evaluation (IITP) grant (2021-0-00537, 2019-0-00075).

---

[5]https://www.fda.gov/Drugs/

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
