# A  Appendix

## A.1  Proofs

### A.1.1  Proof of Proposition 1 (Section 3.2)

We start with the following lemmas:

**Lemma 1.** *Let $\mu$ an equivalence class of order-$(k+l)$ multi-indices. Then, the set of all $\mathbf{i} \in [n]^k$ such that $(\mathbf{i}, \mathbf{j}) \in \mu$ for some $\mathbf{j} \in [n]^l$ is an equivalence class of order-$k$ multi-indices. Likewise, the set of all $\mathbf{j}$ such that $(\mathbf{i}, \mathbf{j}) \in \mu$ for some $\mathbf{i}$ is an equivalence class of order-$l$ multi-indices.*

*Proof.* We only prove for $\mathbf{i}$, as proof for $\mathbf{j}$ is analogous. For some $(\mathbf{i}_1, \mathbf{j}_1) \in \mu$, let us denote $\mathbf{i}_1$'s equivalence class as $\mu_k$. It is sufficient that we prove $\mathbf{i} \in \mu_k \Leftrightarrow (\mathbf{i}, \mathbf{j}) \in \mu$ for some $\mathbf{j}$.

($\Rightarrow$) For all $\mathbf{i} \in \mu_k$, as $\mathbf{i}_1 \sim \mathbf{i}$ we have $\mathbf{i} = \pi(\mathbf{i}_1)$ for some $\pi \in S_n$. As $\pi$ acts on multi-indices entry-wise, we have $\pi(\mathbf{i}_1, \mathbf{j}_1) = (\mathbf{i}, \pi(\mathbf{j}_1))$. As the equivalence pattern is invariant to node permutation by definition, we have $\pi(\mathbf{i}_1, \mathbf{j}_1) \sim (\mathbf{i}, \pi(\mathbf{j}_1)) \sim (\mathbf{i}_1, \mathbf{j}_1)$, and thus $(\mathbf{i}, \pi(\mathbf{j}_1)) \in \mu$. Therefore, for all $\mathbf{i} \in \mu_k$, we always have $(\mathbf{i}, \mathbf{j}) \in \mu$ when we set $\mathbf{j} = \pi(\mathbf{j}_1)$.

($\Leftarrow$) For all $(\mathbf{i}, \mathbf{j}) \in \mu$, as $(\mathbf{i}, \mathbf{j}) \sim (\mathbf{i}_1, \mathbf{j}_1)$ we have $(\mathbf{i}, \mathbf{j}) = \pi(\mathbf{i}_1, \mathbf{j}_1)$ for some $\pi \in S_n$. We have equivalently $\mathbf{i} = \pi(\mathbf{i}_1)$ and $\mathbf{j} = \pi(\mathbf{j}_1)$ for the $\pi$, which leads to $\mathbf{i} \sim \mathbf{i}_1$ and therefore $\mathbf{i} \in \mu_k$.  $\square$

**Lemma 2.** *Let $\mu$ an equivalence class of order-$k$ multi-indices. Then, every $\mathbf{i} \in \mu$ contains the same number of unique elements, which is equal to $|\mu|$ i.e., the number of nonempty subsets in $\mu$'s partition.*

*Proof.* All $\mathbf{i} \in \mu$ have the same equality pattern, specified by $\mu$'s representative partition. Specifically, for all $\mathbf{i} \in \mu$, $\mathbf{i}_a = \mathbf{i}_b$ holds iff $\mathbf{i}_a$ and $\mathbf{i}_b$ belong to the same subset within $\mu$'s partition. Therefore, each nonempty subset within $\mu$'s partition specifies exactly one value within $\mathbf{i}$, and any $\mathbf{i}_a, \mathbf{i}_b$ s.t. $\mathbf{i}_a \neq \mathbf{i}_b$ are contained in distinct subsets within $\mu$'s partition. Thus, each subset in $\mu$'s partition specifies one unique element in $\mathbf{i}$, and we have the number of unique elements in $\mathbf{i}$ equal to $|\mu|$ for all $\mathbf{i} \in \mu$.  $\square$

Now, we prove Proposition 1.

*Proof.* From Lemma 1, let us denote the set of all $\mathbf{i} \in [n]^k$ such that $(\mathbf{i}, \mathbf{j}) \in \mu$ as an order-$k$ equivalence class $\mu_k$, and denote the set of all $\mathbf{j}$ such that $(\mathbf{i}, \mathbf{j}) \in \mu$ as an order-$l$ equivalence class $\mu_q$.

Then, in Eq. (12), to compute $\boldsymbol{\alpha}_{\mathbf{i},\mathbf{j}}^\mu \ \forall (\mathbf{i}, \mathbf{j}) \in \mu$ it is sufficient that we have $\mathbf{K}_{\mathbf{i}}^\mu \ \forall \mathbf{i} \in \mu_k$ and $\mathbf{Q}_{\mathbf{j}}^\mu \ \forall \mathbf{j} \in \mu_q$. Based on the fact, we now analyze and reduce $\mathbf{Q}^\mu = L_{k \to l}^\mu(\mathbf{A})$ ($\mathbf{K}^\mu = L_{k \to k}^\mu(\mathbf{A})$ can be reduced analogously by letting $l = k$). From Eq. (1) and Eq. (2), we can write the computation of $\mathbf{Q}^\mu$ as follows, with $\alpha, \lambda$ equivalence classes of order-$(k+l)$ and order-$l$ multi-indices and $\mathbf{k} \in [n]^k$:

$$\mathbf{Q}_{\mathbf{j}}^\mu = \sum_\alpha \sum_{\mathbf{k}} \mathbf{B}_{\mathbf{k},\mathbf{j}}^\alpha \mathbf{A}_{\mathbf{k}} w_\alpha + \sum_\lambda \mathbf{C}_{\mathbf{j}}^\lambda b_\lambda, \tag{22}$$

$$\text{where} \quad \mathbf{B}_{\mathbf{i},\mathbf{j}}^\alpha = \begin{cases} 1 & (\mathbf{k},\mathbf{j}) \in \alpha \\ 0 & \text{otherwise} \end{cases} ; \quad \mathbf{C}_{\mathbf{j}}^\lambda = \begin{cases} 1 & \mathbf{j} \in \lambda \\ 0 & \text{otherwise} \end{cases} \tag{23}$$

A key idea is that, when we want $\mathbf{Q}_{\mathbf{j}}^\mu$ only for $\mathbf{j} \in \mu_q$, only a subset of equivalence classes among $\alpha$ or $\lambda$ does effective computation and we can discard the rest. Specifically, we can discard an equivalence class $\alpha$ if it contains some $(\mathbf{k},\mathbf{j})$ with $\mathbf{j} \notin \mu_q$. This is because, for such $\alpha$, $(\mathbf{k},\mathbf{j}) \notin \alpha$ if $\mathbf{j} \in \mu_q$, leading to $\mathbf{B}_{\mathbf{k},\mathbf{j}}^\alpha = 0$ if $\mathbf{j} \in \mu_q$. Therefore, such $\alpha$ does not contribute to $\mathbf{Q}_{\mathbf{j}}^\mu \ \forall \mathbf{j} \in \mu_q$ and can be discarded. On the other hand, an equivalence class $\alpha$ containing some $(\mathbf{k},\mathbf{j})$ with $\mathbf{j} \in \mu_q$ does effective computation and should be kept.

From that, it turns out that the number of effective $\alpha$ is $\leq \mathrm{b}(k + u_q)$, where $u_q = u(\mathbf{j}) = |\mu_q|$ is the number of unique entries within some $\mathbf{j} \in \mu_q$ (see Lemma 2). Recall that for an effective $\alpha$, $\mathbf{j} \in \mu_q$ holds for all $(\mathbf{k},\mathbf{j}) \in \alpha$. Within $\alpha$'s representative partition, as each $\mathbf{j} \in \mu_q$ has exactly $u_q$ unique values, we always have $\{\mathbf{j}_1, ..., \mathbf{j}_l\}$ contained in exactly $u_q$ distinct subsets. Thus, the possible number of effective $\alpha$ is upper-bounded by the number of ways of partitioning a set with $k + |\mu_q|$ elements, which is $\mathrm{b}(k + u_q)$. As for the bias, we can repeat the analysis with $k = 0$ and the number of effective $\lambda$ is $\leq \mathrm{b}(u_q)$.

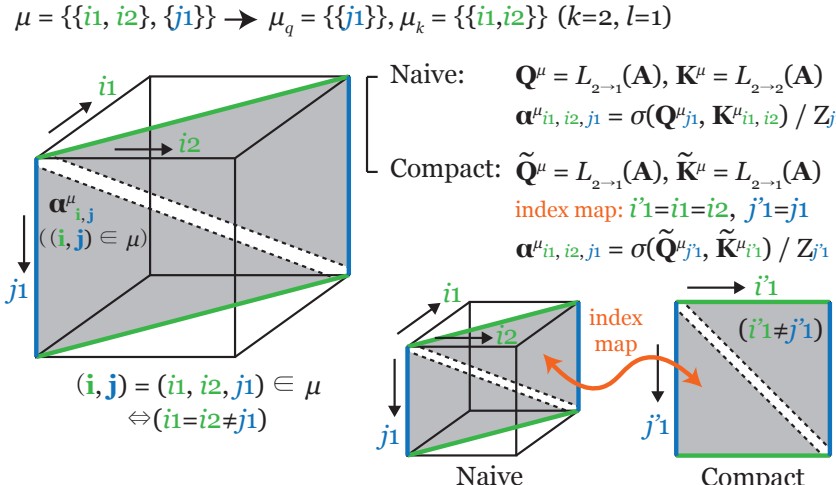

Figure 4: Exemplar illustration of computing $\boldsymbol{\alpha}_{\mathbf{i,j}}^{\mu}\ \forall(\mathbf{i},\mathbf{j}) \in \mu$ with lower-order query and key, for $k = 2, l = 1, \mathbf{i} = (i_1, i_2), \mathbf{j} = (j_1)$, and $\mu = \{\{i_1, i_2\}, \{j_1\}\}$.

We now show that a lower-order linear layer $L_{k\to u_q}^{\mu}$ can compute $\mathbf{Q}_{\mathbf{j}}^{\mu}\ \forall \mathbf{j} \in \mu_q$ in Eq. (22). Let us denote $\mathcal{A}$ the set of all effective $\alpha$ and $\mathcal{L}$ the set of all effective $\lambda$. Then we can rewrite Eq. (22) as:

$$\mathbf{Q}_{\mathbf{j}}^{\mu} = \sum_{\alpha \in \mathcal{A}} \sum_{\mathbf{k}} \mathbf{B}_{\mathbf{k},\mathbf{j}}^{\alpha} \mathbf{A}_{\mathbf{k}} w_{\alpha} + \sum_{\lambda \in \mathcal{L}} \mathbf{C}_{\mathbf{j}}^{\lambda} b_{\lambda}, \tag{24}$$

where $\mathcal{A}$ has $\leq \mathrm{b}(k + u_q)$ elements and $\mathcal{L}$ has $\leq \mathrm{b}(u_q)$ elements. Assume we have some linear layer $L_{k\to u_q}^{\mu}$. With $\mathbf{j}' \in [n]^{u_q}$, and $\beta, \theta$ equivalence classes of order-$(k + u_q)$ and order-$u_q$ multi-indices respectively, we can write:

$$\tilde{\mathbf{Q}}_{\mathbf{j}'}^{\mu} = \sum_{\beta} \sum_{\mathbf{k}} \mathbf{B}_{\mathbf{k},\mathbf{j}'}^{\beta} \mathbf{A}_{\mathbf{k}} w_{\beta} + \sum_{\theta} \mathbf{C}_{\mathbf{j}'}^{\theta} b_{\theta}, \tag{25}$$

$$\text{where}\quad \mathbf{B}_{\mathbf{k},\mathbf{j}'}^{\beta} = \begin{cases} 1 & (\mathbf{k}, \mathbf{j}') \in \beta \\ 0 & \text{otherwise} \end{cases} ; \quad \mathbf{C}_{\mathbf{j}'}^{\theta} = \begin{cases} 1 & \mathbf{j}' \in \theta \\ 0 & \text{otherwise} \end{cases} \tag{26}$$

We now identify the condition that $\tilde{\mathbf{Q}}^{\mu}$ contains all $\mathbf{Q}_{\mathbf{j}}^{\mu}\ \forall \mathbf{j} \in \mu_q$. For that, we need to define a mapping between index space of $\tilde{\mathbf{Q}}^{\mu}$ and $\mathbf{Q}^{\mu}$. To this end, we define a surjection $g : [k] \to [u_q]$ that satisfies $\mathbf{j}_a = \mathbf{j}_b \Leftrightarrow g(a) = g(b)$. We can always define such $g$ due to the property of equivalence classes that, for all $a, b \in [l]$, $\mathbf{j}_a = \mathbf{j}_b$ holds iff $\mathbf{j}_a, \mathbf{j}_b$ belong to a same subset within $\mu_q$'s partition. By indexing the subsets within $\mu_q$'s partition, we define $g(a)\ \forall a \in [l]$ as the index of the subset that $a$ belongs. Then, for every $\mathbf{j} \in \mu_q$, we can find an order-$u_q$ compact form $\mathbf{j}' \in [n]^{u_q}$ containing $u_q$ unique elements through $g$: for $c \in [u_q]$ that $c = g(a) = g(b) = \cdots$, we construct $\mathbf{j}'$ such that $\mathbf{j}_c' = \mathbf{j}_a = \mathbf{j}_b = \cdots$. We define $f_{\mu}^q : [n]^q \to [n]^{u_q}$ as a mapping that gives $\mathbf{j}' = f_{\mu}^q(\mathbf{j})\ \forall \mathbf{j} \in \mu_q$.

We now reduce Eq. (24) into Eq. (25). First, for each $\alpha \in \mathcal{A}$, we assign a distinct order-$(k + u_q)$ equivalence class $\beta$ that satisfies: $(\mathbf{k}, \mathbf{j}) \in \alpha \Leftrightarrow (\mathbf{k}, \mathbf{j}') \in \beta$ with $\mathbf{j}' = f_{\mu}^q(\mathbf{j})$. This can be done by changing $\alpha$'s partition into $\beta$'s, by merging each set of $\mathbf{j}_a = \mathbf{j}_b = ...$ into corresponding $\mathbf{j}_c'$ following $f_{\mu}^q$. We similarly assign a distinct $\theta$ to each $\lambda \in \mathcal{L}$. Then, we set $w_{\alpha} = w_{\beta}$ for all paired $\alpha$ and $\beta$, and set $w_{\beta} = 0$ for every $\beta$ not paired with any $\alpha$. We similarly set $b_{\theta} = b_{\lambda}$ for all paired $\theta$ and $\lambda$, and $b_{\theta} = 0$ for every $\theta$ not paired with any $\lambda$. From the definition of basis tensors, $\mathbf{B}_{\mathbf{k},\mathbf{j}}^{\alpha} = \mathbf{B}_{\mathbf{k},\mathbf{j}'}^{\beta}$ for all paired $\alpha$ and $\beta$, and $\mathbf{C}_{\mathbf{j}}^{\lambda} = \mathbf{C}_{\mathbf{j}'}^{\theta}$ for all paired $\lambda$ and $\theta$. Therefore, we have $\tilde{\mathbf{Q}}_{\mathbf{j}'}^{\mu} = \mathbf{Q}_{\mathbf{j}}^{\mu}$ for all $\mathbf{j} \in \mu_q$. Conclusively, we can always construct $L_{k\to u_q}^{\mu}$ and $f_q^{\mu} : [n]^l \to [n]^{u_q}$ that gives $\mathbf{Q}_{\mathbf{j}}^{\mu}\ \forall \mathbf{j} \in \mu_q$.

As noted in the beginning of the proof, we can perform the same analysis with $k = l$ to show the analogous result for $\mathbf{K}^{\mu}$. We now have all entries $\mathbf{K}_{\mathbf{i}}^{\mu}\ \forall \mathbf{i} \in \mu_k$ and $\mathbf{Q}_{\mathbf{j}}^{\mu}\ \forall \mathbf{j} \in \mu_q$ to compute $\boldsymbol{\alpha}_{\mathbf{i,j}}^{\mu}\ \forall(\mathbf{i},\mathbf{j}) \in \mu$ (Eq. (12)) and therefore Proposition 1 holds. $\qquad\square$

In Figure 4 we provide an example of computing $\boldsymbol{\alpha}^{\mu}$ with lower-order query and key.

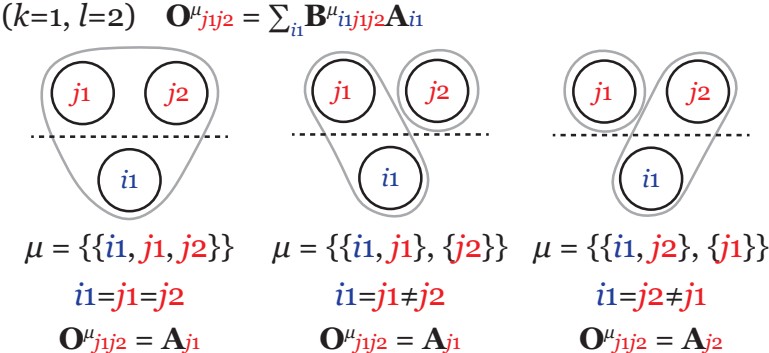

$$(k{=}1,\, l{=}2) \quad \mathbf{O}^\mu_{j1j2} = \sum_{i1} \mathbf{B}^\mu_{i1j1j2} \mathbf{A}_{i1}$$

$\mu = \{\{i1, j1, j2\}\}$  $\quad \mu = \{\{i1, j1\}, \{j2\}\}$  $\quad \mu = \{\{i1, j2\}, \{j1\}\}$

$i1{=}j1{=}j2$ $\qquad\qquad i1{=}j1{\neq}j2$ $\qquad\qquad i1{=}j2{\neq}j1$

$\mathbf{O}^\mu_{j1j2} = \mathbf{A}_{j1}$ $\qquad\quad \mathbf{O}^\mu_{j1j2} = \mathbf{A}_{j1}$ $\qquad\quad \mathbf{O}^\mu_{j1j2} = \mathbf{A}_{j2}$

Figure 5: Exemplar illustration of all equivalence classes included in lightweight linear layer $\bar{L}_{1\to2}$.

### A.1.2 Proof of Property 1 (Section 4)

*Proof.* We begin by analyzing the complexity of an equivariant linear layer $L_{k\to l}$ (Eq. (1)). In the inner summation $\sum_{\mathbf{i}} \mathbf{B}^\mu_{\mathbf{i},\mathbf{j}} \mathbf{A}_{\mathbf{i}}$, with $u_k$ and $u_q$ the number of unique entries in $\mathbf{i}$ and $\mathbf{j}$ respectively, we have $n^{\leq u_k}$ effective multiplication and summations for each output index $\mathbf{j}$. Inequality is when $\mathbf{j}_b = \mathbf{i}_a$ for some $a, b$, which corresponds to indexing operation rather than summation. Thus, the number of operations done by the summation is $n^{u_q} n^{\leq u_k}$. With outer summation over $\mu$, we have $\sum_\mu n^{u_q} n^{\leq u_k}$ operations. As $u_q \leq l$ and $u_k \leq k$ by definition, we have inequality $\sum_\mu n^{u_q} n^{\leq u_k} \leq \mathsf{b}(k+l) n^{k+l}$. Application of $w_\mu$ gives us $dd' \sum_\mu n^{u_q} n^{u_k} \leq \mathsf{b}(k+l) dd' n^{k+l}$ number of operations. For the bias, in the inner term $\mathbf{C}^\lambda_{\mathbf{j}}$, we need a single addition for each $\mathbf{j}$ and thus the number of operations for a $\lambda$ is $n^{u_q} \leq n^l$. Summation over $\lambda$ and application of bias parameters gives us $\mathsf{b}(l) d' n^{u_q} \leq \mathsf{b}(l) d' n^l$ operations. Collectively, we have $\leq \mathsf{b}(k+l) dd' n^{k+l} + \mathsf{b}(l) d' n^l$ number of operations. As $k, l, d, d'$ are constants that does not depend on $n$, we obtain $\mathcal{O}(n^{k+l})$ complexity.

Computation of $\mathrm{Enc}_{k\to l}(\mathbf{A})$ (Eq. (9)) involves computing $\mathrm{Attn}_{k\to l}(\mathbf{A})$, $\mathrm{MLP}_{l\to l}(\mathrm{Attn}_{k\to l}(\mathbf{A}))$ and adding them. Let us analyze $\mathrm{Attn}_{k\to l}(\mathbf{A})$ first. To compute $\boldsymbol{\alpha}^{h,\mu}$ from input, we need to compute $L^\mu_{k\to u_q}(\mathbf{A})$ and $L^\mu_{k\to u_k}(\mathbf{A})$, followed by pairwise similarity computation and re-indexing. Assuming that each pairwise similarity computation and indexing has constant complexity, we have $\mathcal{O}(n^{k+u_q} + n^{k+u_k} + n^{u_q+u_k})$.[6] As $u_q \leq l$ and $u_k \leq k$, we have $\mathcal{O}(n^{k+l} + n^{2k})$. It is worth to note that $\mathcal{O}(n^{2k})$ term comes from computation of keys from input. Having computed $\boldsymbol{\alpha}^{h,\mu}$, the inner summation $\sum_{\mathbf{i}} \boldsymbol{\alpha}^{h,\mu}_{\mathbf{i},\mathbf{j}} \mathbf{A}_{\mathbf{i}}$, similar to in $L_{k\to l}$, has $n^{u_q} n^{\leq u_k}$ computations. With outer summation over $\mu$, we have $\sum_\mu n^{u_q} n^{\leq u_k} \leq \mathsf{b}(k+l) n^{k+l}$ operations. Summation over heads and application of weight matrices gives us $\leq \mathsf{b}(k+l) H d_H^2 dn^{k+l}$ operations, which is $\mathcal{O}(n^{k+l})$. For application of $\mathrm{MLP}_{l\to l}(\cdot)$, we sum the complexity of two linear layers $L_{l\to l}$ and element-wise ReLU, which gives us $\mathcal{O}(n^{2l})$. Conclusively, the complexity of $\mathrm{Enc}_{k\to l}$ is $\mathcal{O}(n^{2k} + n^{k+l} + n^{2l})$. $\qquad\square$

### A.1.3 Proof of Proposition 2 (Section 4.1)

*Proof.* We assume $k, l > 0$. Among the equivalence classes $\mu$ of order-$(k+l)$ multi-indices, let us select a subset $\mathcal{M}$ that all $\mu \in \mathcal{M}$ satisfies the following: for all $(\mathbf{i}, \mathbf{j}) \in \mu$, $\mathbf{i}_a = \mathbf{j}_b$ holds for all $a \in [k]$ and some $b \in [l]$. In other words, every element in $\mathbf{i}$ is identical with at least one element in $\mathbf{j}$, and $\mathbf{i}$ becomes a single fixed multi-index when we fix $\mathbf{j}$ (we denote the fixed $\mathbf{i} = \mathrm{fix}(\mathbf{j})$). This renders $\mathbf{B}^\mu_{\mathbf{i},\mathbf{j}} = 1 \Leftrightarrow \mathbf{i} = \mathrm{fix}(\mathbf{j})$ for such $\mu$, and consequently the inner-summation $\sum_{\mathbf{i}} \mathbf{B}^\mu_{\mathbf{i},\mathbf{j}} \mathbf{A}_{\mathbf{i}} w_\mu$ in Eq. (14) reduces to elementwise indexing $\mathbf{A}_{\mathrm{fix}(\mathbf{j})} w_\mu$. As the size of $\mathcal{M}$ is upper-bounded by a constant $\mathsf{b}(k+l)$, we have $\mathcal{O}(n^l)$ complexity when computing Eq. (14). With the trivial case $\mu = \{\{i_1, ..., i_k, j_1, ..., j_l\}\} \in \mathcal{M}$, we can always find nonempty $\mathcal{M}$. $\qquad\square$

To provide some intuition, we illustrate all $\mu \in \mathcal{M}$ for $k = 1, l = 2$ in Figure 5.

---

[6]Normalization over keys gives an additive complexity $\mathcal{O}(n^{u_q+u_k})$, which can be absorbed to the formula.

### A.1.4 Validity of Proposition 1 when using lightweight linear layers (Section 4.1)

As stated in the main text, $\text{Enc}_{k\to l}$ (Eq. (9)) with linear layers for key, query, and MLP changed to $\bar{L}$ still generalizes $L_{k\to l}$. This can be shown simply by plugging $\bar{L}$ into the proof of Proposition 1. We can still assume $\boldsymbol{\alpha}_{\mathbf{i},\mathbf{j}}^{h,\mu} = 1$ for all $(\mathbf{i},\mathbf{j}) \in \mu$ by setting $\bar{L}$ for key and query to output constants, and can reduce $\text{MLP}_{l\to l}$ composed of $\bar{L}$ to an invariant bias as we subsample $\mu \in \mathcal{M}$ but keep all $\lambda$ for the bias. Thus, Eq. (9) can reduce to Eq. (1) and Proposition 1 holds.

### A.1.5 Proof of Property 2 (Section 4.2)

*Proof.* We begin from sparse equivariant linear layer $L_{k\to l}$ (Eq. (15)). In the inner summation $\sum_{\mathbf{i}\in E} \mathbf{B}_{\mathbf{i},\mathbf{j}}^{\mu} \mathbf{A}_{\mathbf{i}}$, the number of multiplication and addition for each $\mathbf{j}$ is upper-bounded by $m = |E|$. As the number of output multi-indices $\mathbf{j}$ is bounded by $|E'| \leq m\binom{k}{l}$, the effective number of operations are $\leq m^2 \binom{k}{l}$. With outer summation over $\mu$, we have $\leq \mathrm{b}(k+l)\binom{k}{l}m^2$ operations, leading to complexity $\mathcal{O}(m^2)$. For the lightweight linear layers $\bar{L}$ (Proposition 2) that precludes summation over input, we trivially have $\mathcal{O}(m)$ complexity as we do not sum over $\mathbf{i}$.

Now, we analyze the complexity of sparse self-attention computation (Eq. (16)). To compute $\boldsymbol{\alpha}^{\mu}$ from input, we need to compute lightweight linear layers $\bar{L}_{k\to u_q}^{\mu}(\mathbf{A}, E)$ and $\bar{L}_{k\to u_k}^{\mu}(\mathbf{A}, E)$, followed by pairwise similarity computation of nonzero entries. As $u_q, u_k \leq k$, we have complexity $\mathcal{O}(m)$ for the linear layers and $\mathcal{O}(m^2)$ for pairwise computation. Having computed $\boldsymbol{\alpha}^{\mu}$, the inner summation $\sum_{\mathbf{i}\in E} \boldsymbol{\alpha}_{\mathbf{i},\mathbf{j}}^{\mu} \mathbf{A}_{\mathbf{i}}$ has $\leq m$ computations. Enumerating over $\mathbf{j}$, we have $\mathcal{O}(m^2)$.

Finally, we analyze the complexity of $\text{Enc}_{k\to l}$ composed of the sparse linear layers and self-attention. This involves adding the outputs of $\text{Attn}_{k\to l}(\mathbf{A}, E)$ (which we already addressed) and $\text{MLP}_{l\to l}(\text{Attn}_{k\to l}(\mathbf{A}, E), E)$. For application of $\text{MLP}_{l\to l}$, we sum the complexity of two lightweight linear layers $\bar{L}_{l\to l}$ and element-wise ReLU, which gives us $\mathcal{O}(m)$. In summary, the complexity of sparse $\text{Enc}_{k\to l}$ is $\mathcal{O}(m^2)$. $\square$

### A.1.6 Proof of Property 3 (Section 4.3)

*Proof.* Summation over $\mathbf{i} \in \mathcal{I}$ decouples $\mathbf{i}$ from $\mathbf{j}$ and allows reuse of computation over $\mathbf{j}$. As the summation over $\mathbf{i} \in \mathcal{I}$ involves $\mathcal{O}(n^k)$ operations and we share it over all query indices $\mathbf{j}$, self-attention reduces to elementwise application and we obtain $\mathcal{O}(n^k + n^l)$ complexity. As computation of $\tilde{\mathbf{Q}}^{\mu}, \tilde{\mathbf{K}}^{\mu}$ and application of $\text{MLP}_{l\to l}$ are $\mathcal{O}(n^k + n^l)$ with lightweight linear layers, we have $\mathcal{O}(n^k + n^l)$ collective complexity for $\text{Enc}_{k\to l}$. When adopted into sparse $\text{Enc}_{k\to l}$ (Eq. (16)), summation over $\mathbf{i}$ and enumeration over $\mathbf{j}$ all reduce to $\mathcal{O}(m)$ and we thereby have $\mathcal{O}(m)$ complexity. $\square$

### A.1.7 Proof of Theorem 2 (Section 4.4)

*Proof.* With message function $M : \mathbb{R}^{2d_v + d_e} \to \mathbb{R}^{d_m}$ and update function $U : \mathbb{R}^{d_v + d_m} \to \mathbb{R}^d$, a message passing step takes node features $\mathbf{X} \in \mathbb{R}^{n \times d_v}$ and edge features $\mathbf{E} \in \mathbb{R}^{n \times n \times d_e}$ as input and outputs node features $\mathbf{H} \in \mathbb{R}^{n \times d}$ according to following. With $i, j \in [n]$:

$$\mathbf{M}_j = \sum_{i \in \mathcal{N}(j)} M(\mathbf{X}_j, \mathbf{X}_i, \mathbf{E}_{ij}) \tag{27}$$

$$\mathbf{H}_j = U(\mathbf{X}_j, \mathbf{M}_j), \tag{28}$$

where $\mathcal{N}(j)$ denotes incoming neighbors of $j$-th node, *i.e.,* $\{i | (i,j) \in E\}$.

We now show how a composition of two $\text{Enc}_{2\to 2}$ can approximate above computation.

1. As a first step, we encode $\mathbf{X}$ and $\mathbf{E}$ into a single $\mathbf{A} \in \mathbb{R}^{n \times n \times (2d_v + d_e)}$ [26]. In the first $d_v$ channels, we replicate $\mathbf{X}$ on the rows. In the next $d_v$ channels, we replicate $\mathbf{X}$ on the columns. In the last $d_e$ channels, we put $\mathbf{E}$. Additionally, to account for output positions (node features), we augment $E$ with self-loops and make $E' = E \cup \{(i,i)\ \forall i \in [n]\}$.

2. Then, we make the first $\text{Enc}_{2\to 2}$ approximate the message function $M(\cdot)$, so that $\text{Enc}_{2\to 2}(\mathbf{A})_{ij} \approx M(\mathbf{X}_j, \mathbf{X}_i, \mathbf{E}_{ij})$. To do this, we first reduce $\text{Attn}_{2\to 2}(\mathbf{A})_{ij} = \mathbf{A}_{ij}$ and apply $\text{MLP}_{l\to l}$ on top of it. We reduce $\text{MLP}_{l\to l}$ to entry-wise MLP. As $\text{Attn}_{2\to 2}(\mathbf{A})_{ij} = \mathbf{A}_{ij}$

is a concatenation of $\mathbf{X}_i, \mathbf{X}_j, \mathbf{E}_{ij}$, with universal approximation theorem [16], we can have the output of the first $\text{Enc}_{2\to2}(\mathbf{A})_{ij} = M(\mathbf{X}_j, \mathbf{X}_i, \mathbf{E}_{ij}) + \epsilon_1 \; \forall i, j$ where $\epsilon_1$ is approximation error. This also holds when we leverage sparsity and restrict the index scope to $(i, j) \in E'$; in this case, we make $(i, j) \in E' \setminus E$ contain zero vectors.

3. Before feeding the output to the second $\text{Enc}_{2\to2}$, we concatenate the original input $\mathbf{A}$ with the output of the first layer in the channel dimension to make $\mathbf{A}' \in \mathbb{R}^{n \times n \times (2d_v + d_e + d_m)}$. This gives $\mathbf{X}_i, \mathbf{X}_j, \mathbf{E}_{ij}$ encoded in the first $2d_v + d_e$ channels and $M(\mathbf{X}_j, \mathbf{X}_i, \mathbf{E}_{ij}) + \epsilon_1$ encoded in the last $d_m$ channels of $\mathbf{A}'$. This operation can be trivially absorbed within the MLP of the first layer, but we separate for simplicity.

4. Now, we make the second $\text{Enc}_{2\to2}$ jointly approximate summation of messages over neighbors $\sum_{i \in \mathcal{N}(j)}(\cdot)$ and update function $U(\cdot)$, so that $\text{Enc}_{2\to2}(\mathbf{A}')_{jj} \approx \mathbf{H}_j = U(\mathbf{X}_j, \mathbf{M}_j)$. First, we reduce $\text{Attn}_{2\to2}(\mathbf{A}')$ to summation over neighbors. For this we only need two equivalence classes $\mu_1 = \{\{1\}, \{2, 3, 4\}\}$ and $\mu_2 = \{\{1, 2, 3, 4\}\}$. Omitting normalization, we can write Eq. (21) as follows. For $\mu_1$ we set $u_q = 1$, $u_k = 2$, $\mathbf{i} = (i, j)$, $\mathbf{j} = (j, j)$, $\mathbf{i}' = (i, j)$, $\mathbf{j}' = j$, and for $\mu_2$ we set $u_q = 1$, $u_k = 1$, $\mathbf{i} = (j, j)$, $\mathbf{j} = (j, j)$, $\mathbf{i}' = j$, $\mathbf{j}' = j$.

$$\text{Attn}_{2\to2}(\mathbf{A}')_{jj} = \phi(\tilde{\mathbf{Q}}_j^{\mu_1})^\top \sum_{\{i|(i,j)\in E'\}} \phi(\tilde{\mathbf{K}}_{ij}^{\mu_1})\mathbf{A}'_{ij}w_{\mu_1} + \phi(\tilde{\mathbf{Q}}_j^{\mu_2})^\top \phi(\tilde{\mathbf{K}}_j^{\mu_2})\mathbf{A}'_{jj}w_{\mu_2}.$$

(29)

Let entries in $\phi(\tilde{\mathbf{K}}^\mu), \phi(\tilde{\mathbf{Q}}^\mu)$ be $\frac{1_{d_K}}{\sqrt{d_K}}$ so that their dot product is 1. Eq. (29) reduces to:

$$\text{Attn}_{2\to2}(\mathbf{A}')_{jj} = \sum_{\{i|(i,j)\in E'\}} \mathbf{A}'_{ij}w_{\mu_1} + \mathbf{A}'_{jj}w_{\mu_2}$$

(30)

We make $w_{\mu_1}$ zero-out the first $2d_v + d_e$ channels and $w_{\mu_2}$ zero-out the last $d_e + d_m$ channels. Then, we have $\text{Attn}_{2\to2}(\mathbf{A}')_{jj}$ contain $\mathbf{X}_j$ in the first $d_v$ channels and $\mathbf{M}_j + \epsilon_2 = \sum_{i|(i,j)\in E'}(M(\mathbf{X}_j, \mathbf{X}_i, \mathbf{E}_{ij})) + \epsilon_2$ in the last $d_m$ channels where $\epsilon_2$ is approximation error[7]. We then apply $\text{MLP}_{l\to l}$ on top of it, which can approximate the update function by universal approximation theorem [16] and we have $\text{Attn}_{2\to2}(\mathbf{A}')_{jj} = U(\mathbf{X}_j, \mathbf{M}_j) + \epsilon_3$ where $\epsilon_3$ is approximation error.

Overall, the approximation error $\epsilon_i$ at each step depends on $\epsilon_{i-1}$ $(i > 1)$, the MLP that approximates relevant function, and uniform bounds and uniform continuity of the approximated functions [16].

In the opposite, message passing cannot approximate some of the operations done by a single Transformer layer $\text{Enc}_{2\to2}$. This can be seen from the fact that, given a graph with diameter $d(E)$, we need at least $d(E)$ message passing operations to approximate output of $\text{Enc}_{2\to2}$. This is because a single $\text{Enc}_{2\to2}$ can impose dependency between any pair of input and output indices $\mathbf{i}, \mathbf{j}$, while message passing requires $d(E)$ steps in the worst case. Consequently, the approximation becomes impossible when the graph contains $> 1$ disconnected components, which leads to $d(E) \to +\infty$. $\square$

## A.2 Experimental details (Section 5)

In this section, we provide detailed information of the datasets and models used in our experiments in Section 5. We provide the dataset statistics in Table 5, and model architectures in Table 6.

### A.2.1 Implementation details of higher-order Transformers

In formulation of higher-order Transformers in the main text, for simplicity we omitted layer normalization (LN) [1] and used ReLU non-linearity for $\text{MLP}_{l\to l}$. In actual implementation, we adopt Pre-Layer Normalization (PreLN) [33], and place layer normalization before $\text{Attn}_{k\to l}$, before $\text{MLP}_{l\to l}$, and before the output linear projection after the last $\text{Enc}_{k\to l}$. We also use GeLU non-linearity [15] in $\text{MLP}_{l\to l}$ instead of ReLU. This setup worked robustly in all experiments. As additional details, we set the internal dimension of $\text{MLP}_{l\to l}$ same as the input and output dimension ($d_F = d$), and applied dropout [30] within $\text{Attn}_{k\to l}$ and $\text{MLP}_{l\to l}$ to prevent overfitting.

---

[7]Note that message summation over $\{i|(i, j) \in E'\}$ is equivalent to summation over $\{i|(i, j) \in E\} = \mathcal{N}(j)$ because we set message zero at $(i, j) \in E' \setminus E$.

Table 5: Statistics of the datasets.

(a) Statistics of the synthetic chains dataset.

| Dataset | Chains |
|---|---|
| Size | 60 |
| # classes | 2 |
| Average # node | 20 (train) / 200 (test) |

(b) Statistics of the PCQM4M-LSC dataset.

| Dataset | PCQM4M-LSC |
|---|---|
| Size | 3.8M |
| Average # node | 14.1 |
| Average # edge | 14.6 |

(c) Dataset statistics for set-to-graph prediction.

| Dataset | Jets | Delaunay (50) | Delaunay (20-80) |
|---|---|---|---|
| Size | 0.9M | 55k | 55k |
| Average # node | 7.11 | 50 | 50 |
| Average # edge | 35.9 | 273.6 | 273.9 |

(d) Dataset statistics for $k$-uniform hyperedge prediction. For each dataset, each row under "# nodes" correspond to each row under "Node types".

| Dataset | GPS | MovieLens | Drug |
|---|---|---|---|
| Node types | user location activity | user movie tag | user drug reaction |
| # nodes | 146 70 5 | 2,113 5,908 9,079 | 12 1,076 6,398 |
| # edges | 1,436 | 47,957 | 171,756 |

## A.2.2 Efficient implementation of $1 \to k$ layers

For $k$-uniform hyperedge prediction in Sec. 5, implementing the higher-order layers $\mathrm{Enc}_{1 \to k}$ and $L_{1 \to k}$ can be challenging due to the large number of equivalence classes, $b(1+k)$. However, we found that it can be reduced to $1+k$ without any approximation. Specifically, we show the following:

**Property 4.** *For $L_{1 \to k}$ or $\mathrm{Enc}_{1 \to k}$, if we only consider $k$-uniform output hyperedges (output hyperedges without loops; $\mathbf{j}$-th output where $\mathbf{j}_1, ..., \mathbf{j}_k$ are unique), the layers can be implemented using only $1+k$ equivalence classes instead of $b(1+k)$.*

*Proof.* As we only care about output hyperedges with unique index elements, only equivalence classes that correspond to partitions of $[k+1]$ with entries $[k]$ contained in disjoint subsets contribute to output. There are exactly $1+k$ such partitions depending on which subset the last entry $(k+1)$ belongs to, so it is sufficient that we have $1+k$ equivalence classes. $\square$

From Property 4, we implement the layers $\mathrm{Enc}_{1 \to k}$ and $L_{1 \to k}$ by only considering the $1+k$ equivalence classes that contribute to $k$-uniform output hyperedges.

## A.2.3 Runtime and memory analysis

For runtime and memory analysis, we used Barabási-Albert random graphs that are made by iteratively adding nodes, where each added node links to 5 random previous nodes. The experiment was done using a single RTX 6000 GPU with 22GB. We repeated the experiment 10 times with different random seeds for graph generation and reported the average; variance was generally low. The architectures of the experimented second-order models are provided in Table 6a.

## A.2.4 Synthetic chains

For synthetic chains experiment, we used a small dataset composed of 40 training chains each with 20 nodes, and 20 test chains each with 200 nodes as in Table 5a. Each chain is randomly assigned

Table 6: Architectures of the models used in our experiments. $\text{Enc}_{k \to l}(d, d_H, H)$ denotes $\text{Enc}_{k \to l}$ with hidden dimension $d$, head dimension $d_H$, and number of heads $H$. $L_{k \to l}(d)$ denotes $L_{k \to l}$ with output dimension $d$. $\text{MLP}(n, d, d_{\text{out}})$ denotes an elementwise MLP with $n$ hidden layers, hidden dimension $d$, output dimension $d_{\text{out}}$, and ReLU non-linearity.

(a) Architectures for runtime and memory analysis.

| Method | Architecture |
|---|---|
| $\text{MLP}_\pi$ (D/S) | $[L_{2 \to 2}(32) - \text{ReLU}]_{\times 4}$-$L_{2 \to 0}(32)$ |
| Ours (D/S, ($\phi$)) | $\text{Enc}_{2 \to 2, \phi}(32, 8, 4)_{\times 4}$-$\text{Enc}_{2 \to 0}(32, 8, 4)$-LN-Linear(32) |

(b) Architectures for chain experiment. Output dimension of a layer is denoted in parenthesis.

| Method | Architecture |
|---|---|
| GCN | GCNConv(16)-ReLU-GCNConv(16)-Linear(2) |
| GIN-0 | GINConv(16)-ReLU-GINConv(16)-Linear(2) |
| GAT | GATConv(16)-ReLU-GATConv(16)-Linear(2) |
| $\text{MLP}_\pi$ (S) | $L_{2 \to 2}(16)$-ReLU-$L_{2 \to 1}(16)$-Linear(2) |
| Ours (S) w/o global | $\text{Enc}_{2 \to 2, \text{ablated}}(16)$-$\text{Enc}_{2 \to 1, \text{ablated}}(16)$-LN-Linear(2) |
| Ours (S, $\phi$) w/o global | $\text{Enc}_{2 \to 2, \phi, \text{ablated}}(16)$-$\text{Enc}_{2 \to 1, \phi, \text{ablated}}(16)$-LN-Linear(2) |
| Ours (S) | $\text{Enc}_{2 \to 2}(16)$-$\text{Enc}_{2 \to 1}(16)$-LN-Linear(2) |
| Ours (S, $\phi$) | $\text{Enc}_{2 \to 2, \phi}(16)$-$\text{Enc}_{2 \to 1, \phi}(16)$-LN-Linear(2) |

(c) Architectures for graph regression experiment.

| Method | Architecture |
|---|---|
| Transformer + Laplacian PE | $\text{Enc}_{1 \to 1}(256, 16, 16)_{\times 8}$-$\text{Enc}_{1 \to 0}(256, 16, 16)$-LN-Linear(1) |
| $\text{MLP}_\pi$ (S) | $[L_{2 \to 2}(256) - \text{ReLU}]_{\times 8}$-$L_{2 \to 0}(256)$-Linear(1) |
| Ours (S, $\phi$)$_{-\text{SMALL}}$ | $\text{Enc}_{2 \to 2, \phi}(256, 8, 4)_{\times 8}$-$\text{Enc}_{2 \to 0}(256, 16, 8)$-LN-Linear(1) |
| Ours (S, $\phi$) | $\text{Enc}_{2 \to 2, \phi}(512, 16, 4)_{\times 8}$-$\text{Enc}_{2 \to 0}(512, 16, 16)$-LN-Linear(1) |

(d) Architectures for set-to-graph experiment.

| Method | Dataset | Architecture |
|---|---|---|
| S2G/S2G+ [29] | Jets | $[L_{1 \to 1}(256) - \text{ReLU}]_{\times 5}$-$L_{1 \to 2}(256)$-MLP(1, 256, 1) |
| | Delaunay (50) | $[L_{1 \to 1}(500) - \text{ReLU}]_{\times 7}$-$L_{1 \to 2}(500)$-MLP(2, 1000, 1) |
| | Delaunay (20-80) | $[L_{1 \to 1}(500) - \text{ReLU}]_{\times 7}$-$L_{1 \to 2}(500)$-MLP(2, 1000, 1) |
| Ours (D) | Jets | $\text{Enc}_{1 \to 1}(128, 32, 4)_{\times 4}$-$\text{Enc}_{1 \to 2}(128, 32, 4)$-MLP(1, 256, 1) |
| | Delaunay (50) | $\text{Enc}_{1 \to 1}(256, 64, 4)_{\times 5}$-$\text{Enc}_{1 \to 2}(256, 64, 4)$-MLP(2, 256, 1) |
| | Delaunay (20-80) | $\text{Enc}_{1 \to 1}(256, 64, 4)_{\times 5}$-$\text{Enc}_{1 \to 2}(256, 64, 4)$-MLP(2, 256, 1) |
| Ours (D, $\phi$) | Jets | $\text{Enc}_{1 \to 1, \phi}(128, 32, 4)_{\times 4}$-$\text{Enc}_{1 \to 2, \phi}(128, 32, 4)$-MLP(1, 256, 1) |
| | Delaunay (50) | $\text{Enc}_{1 \to 1, \phi}(256, 64, 4)_{\times 5}$-$\text{Enc}_{1 \to 2, \phi}(256, 64, 4)$-MLP(2, 256, 1) |
| | Delaunay (20-80) | $\text{Enc}_{1 \to 1, \phi}(256, 64, 4)_{\times 5}$-$\text{Enc}_{1 \to 2, \phi}(256, 64, 4)$-MLP(2, 256, 1) |

(e) Architectures for $k$-uniform hyperedge prediction experiment.

| Method | Dataset | Architecture |
|---|---|---|
| S2G+ (S) | GPS | $[L_{1 \to 1}(64) - \text{ReLU}]_{\times 1}$-$L_{1 \to 3}(64)$-MLP(4, 64, 1) |
| | MovieLens | $[L_{1 \to 1}(64) - \text{ReLU}]_{\times 3}$-$L_{1 \to 3}(64)$-MLP(2, 64, 1) |
| | Drug | $[L_{1 \to 1}(64) - \text{ReLU}]_{\times 3}$-$L_{1 \to 3}(64)$-MLP(2, 64, 1) |
| Ours (S, $\phi$) | GPS | $\text{Enc}_{1 \to 1, \phi}(64, 16, 8)_{\times 1}$-$\text{Enc}_{1 \to 3, \phi}(64, 16, 8)$-MLP(4, 64, 1) |
| | MovieLens | $\text{Enc}_{1 \to 1, \phi}(64, 16, 8)_{\times 3}$-$\text{Enc}_{1 \to 3, \phi}(64, 16, 8)$-MLP(2, 64, 1) |
| | Drug | $\text{Enc}_{1 \to 1, \phi}(64, 16, 8)_{\times 3}$-$\text{Enc}_{1 \to 3, \phi}(64, 16, 8)$-MLP(2, 64, 1) |

with a binary label, which is encoded as one-hot vector at a terminal node. The goal is to classify all nodes in the chain according to the label. As evaluation metrics, we used macro-/micro-F1 that give combined node-wise F1 scores across all test chains. All models, including baselines, have fixed hyperparameters with 2 layers and 16 hidden dimensions. Detailed architectures are provided in Table 6b. For update function of GIN-0, we used an MLP with of 2 hidden layers followed by batchnorm (Linear(16)-ReLU-Linear(16)-ReLU-BN). For GAT, we used 8 attention heads followed by channelwise sum. For second-order Transformers, we used a simplified architecture with a single attention head. We trained all models with binary cross-entropy loss and Adam optimizer [20] with learning rate 1e-3 and batch size 16 for 100 epochs.

### A.2.5 Large-scale graph regression

For large-scale graph regression, we used the PCQM4M-LSC quantum chemistry regression dataset from OGB-LSC benchmark [17], one of the largest datasets up to date that contains $3.8M$ molecular graphs. We provide the summary statistics of the dataset in Table 5b. As the test set is unavailable, we report and compare the Mean Absolute Error (MAE) measured on the validation set.

Table 6c gives the architectures of the models used in our experiment. For second-order models ($MLP_\pi$ and Ours (S, $\phi$)), we used both node and edge types as input information. For vanilla (first-order) Transformer that operates on node features only, we used Laplacian graph embeddings [2, 8] in addition to node types so that the model can consider edge structure information. The embeddings are computed by factorizing the graph Laplacian matrix [8]:

$$\Delta = I - D^{-1/2}AD^{-1/2} = U^\top \Lambda U, \tag{31}$$

where $A$ is the adjacency matrix, $D$ is the degree matrix, and $\Lambda, U$ are the eigenvalues and eigenvectors respectively. Following prior work [8], we used the $k$ smallest eigenvectors of a node.

We trained all models with L1 loss using AdamW optimizer [24] with batch size 1024 on 8 RTX 3090 GPUs. For all models, we used dropout rate of 0.1 to prevent overfitting. For the full schedule, we trained our model for 1M steps, and applied linear learning rate warm-up [31] for 60k steps up to 2e-4 followed by linear decay to 0. For the short schedule (* in Table 2), we trained the models for 100k steps, and applied learning rate warm-up for 5k steps up to 1e-4 followed by decay to 0.

### A.2.6 Set-to-graph prediction

For set-to-graph prediction experiment, we borrow the datasets, code, and baseline scores from Serviansky et. al. (2020) [29]. We provide the summary statistics of the datasets in Table 5c.

As in main text, Jets is a dataset where the task is to infer partition of a set of observed particles. By viewing each partition as a fully-connected graph, the task becomes graph prediction problem. Each data instance contains 2-14 nodes, each having 10-dimensional features. The entire dataset contains 0.9M instances, divided into 60/20/20% train/val/test sets. Evaluation is done with 3 metrics: F1 score, Rand Index (RI), and Adjusted Rand Index (ARI) which is computed as $ARI = (RI - \mathbb{E}[RI])/(1 - \mathbb{E}[RI])$. To ensure that the model's prediction gives a correct partitioning, a postprocessing is applied to convert every connected components to cliques. The test set is further separated into 3 types: bossom(B)/charm(C)/light(L), depending on underlying data generation process. This makes typical # of partitions in each set different. Among the baselines, GNN is a message-passing GNN [10] that operate on k-NN induced graph for $k = 5$, where edge prediction is done with pairwise dot-product. AVR is an algorithmic baseline typically used in particle physics.

As in main text, Delaunay datasets involve 2D point sets where the task is performing Delaunay triangulation. Evaluation metrics are typical Accuracy/Precision/Recall/F1 scores based on edge-wise binary classification on held-out test set. The baselines are similar to Jets; GNN0/5/10 are message-passing GNNs [10] that operate on k-NN induced graph for $k \in \{0, 5, 10\}$.

Table 6d provides the architecture of the models used in our experiment, along with relevant baselines S2G/S2G+ from Serviansky et. al. (2020) [29]. S2G uses a subset of equivalence classes ($\mu$) within $L_{1\to2}$, and S2G+ uses full basis[8]. Our models, both (D) and (D, $\phi$), are made by substituting $Enc_{1\to1}$ and $Enc_{1\to2}$ into S2G+. All models were trained with Adam optimizer to minimize the combination of soft F1 score and binary cross-entropy of edge prediction. For all models, we used dropout rate

---

[8]Note that the implementation of linear layers in Serviansky et. al. (2020) [29] is slightly different from ours.

of 0.1 to prevent overfitting. For Jets, with 400 max epochs, the training is early-stopped based on validation F1 score with 20-epoch tolerance. We used learning rate 1e-4 and batch size 512 for our models, while S2G/S2G+ used learning rate 1e-3 and batch size 2048 [29]. For Delaunay, with 100 max epochs, we used learning rate 1e-4 for our models, and used batch size 32/16 for Delaunay (50)/(20-80); S2G/S2G+ used learning rate 1e-3 and batch size 64 [29]. For Ours (D, $\phi$) in Delaunay (20-80), we applied 1-epoch warmup to prevent early training instability.

### A.2.7 $k$-uniform hyperedge prediction

For $k$-uniform hyperedge prediction experiment, we borrow the datasets, code, and baseline scores from Zhang et. al. (2020) [36]. As in the main text, we used three datasets for transductive 3-edge prediction. The first dataset GPS contains (user-location-activity) hyperedges. The second dataset MovieLens contains (user-movie-tag) hyperedges. The third dataset Drug contains (user-drug-reaction) hyperedges. We provide the summary statistics in Table 5a.

The experiments were done in a transductive setup, where the hyperedge set is randomly split into the training and test set with 4:1 ratio. We randomly sampled negative edges to be 5 times the amount of the positive edges, so that hyperedge prediction becomes binary classification problem. Thus, the evaluation is done with AUC and AUPR scores. Among the baselines, for Hyper-SAGNN, we reproduced the scores using the open-sourced code [36] using the provided hyperparameters. For additional baselines including node2vec, we take the scores reported in Zhang et. al. (2020) [36].

Table 6e gives the architecture of the models used in our experiment. As Hyper-SAGNN uses autoencoder-based node features, for proper comparison we also adopted and trained them jointly with the full model [36]. All models (including reproduced Hyper-SAGNN) were trained with Adam optimizer to minimize the combination of binary cross-entropy loss and autoencoder reconstruction loss for 300 epochs with learning rate 1e-3 and batch size 96. For S2G+ (S) and Ours (S, $\phi$), we applied dropout rate of 0.1 to the hidden layers of MLP after $L_{1\to3}$ or $Enc_{1\to3}$ to prevent overfitting.

### A.3 Potential negative social impacts

Our framework can be potentially applied to a variety of tasks involving relational data, e.g., molecular structures, social networks, 3D mesh, etc. Advancements in those directions might incur negative side-effects such as low-cost biochemical weapon, deepening of filter bubbles from enhanced personalized social network services, surveillance with mesh-based face recognition, etc. Such potential negative impacts should be addressed as we conduct domain-specific follow-up works.