# OpenReview forum: "Transformers Generalize DeepSets and Can be Extended to Graphs & Hypergraphs"
_NeurIPS.cc/2021/Conference — NeurIPS 2021 Poster_

### Official Review · Reviewer_tk1s · 2021-07-11

**Rating:** 6
**Confidence:** 3

**Summary:**

The authors propose a framework to generalize transformers to any order permutation invariant data, especially hypergraphs. While the generalization itself is incremental compare to [20], the naïve approach gives unbearable computational complexity. Hence, the authors propose a series of tricks to lower the computational complexity which is their main contribution.

**Limitations And Societal Impact:**

Yes

**Main Review:**

Strength:

(+) The idea itself sounds appealing.

(+) The experiment results seem to provide evidence that the proposed method has better performance in general.

Weaknesses:

(-) The proposed tricks to reduce computational complexity are lack clarity. Moreover, some of them seem problematic in simple examples. (See detail comments)

(-) The authors only use $L_{2\rightarrow 2}$ layers at most. Hence, it is unclear to see whether the model complexity is indeed $O(m)$, independent of orders $l,k$.

(-) This paper mainly focuses on the case where the input is a hypergraph. However, neither any experiments on hypergraphs nor hypergraph neural network baselines are included in the experiment section.

Detail comments:

While I like the idea of this paper a lot, its unclarity avoids me from supporting acceptance. My first concern is about its complexity reduction tricks. I find myself have at least one doubt for all three tricks (Section 4.1, 4.2, and 4.3).

In section 4.1, line 191-198. In the example of $L_{1\rightarrow 1}$, the authors mention that they only use the basis $I_n$ and ignore the sum-pooling $11^T$ in Eq (4). Then they claim this light-weighted design is still able to model “interactions” between input elements. However, if we drop the sum-pooling $11^T$ in Eq (4), then no interactions between elements are considered, as now it only uses the basis $I_n$. This part really confuses me. Please elaborate more. Note that in the simple graph example, the proposed light-weighted design of $L_{1\rightarrow 1}$ is to ignore all “edge features” which means the graph information is completely neglected. Clearly, it is impossible to recover this information barely from attention among node features in general.

The heuristic approach of Section 4.2, especially the use of E’ is unclear. Since the authors say it is a heuristic, I expect that this approach will lose some information in (at least) some special cases? Please provide a minimal illustrative example (i.e. k=2, l=1?) to explain the use of E’.

Line 240. The definition of set $\mathcal{I}$ is unclear. Does it mean the set that union all $\mathbf{i},\forall (\mathbf{i},\mathbf{j})\in \mu$ over all possible $\mathbf{j}$? If this is the case, then shouldn’t the term that we actually summing over in eq (19) “less” than eq (20)? Why this resulting in further reduction needs more elaboration.

On the experiment side, since I feel the main contribution of the current paper is reducing the complexity of higher-order Transformer, I think the authors should at least conduct one experiment on it. More precisely, for a fixed hypergraph with $m$ hyperedges, I expect to see the experiment showing the time complexity of the proposed model remain unchanged even for increasing order $l,k$ in $L_{l \rightarrow k}$. I think it is necessary to verify the proposed analysis result by experiments. In line 327-334. The authors mention that due to the growth of $k^{th}$ order bell number, the higher-order model is actually infeasible. In this case, I feel the importance and contribution of the current paper are greatly degraded.

Finally, since the paper mainly focuses on the case where the input is a hypergraph, I think the authors should have some experiments regarding hypergraphs. However, neither any experiments on hypergraphs nor hypergraph neural network baselines are included in the experiment section. I suggest the authors either conduct experiments on hypergraphs and compare with SOTA hypergraph neural networks, or tone down the emphasis of hypergraphs at least in the abstract and introduction.

***
After rebuttal:

I really appreciate the detailed response by the authors, which resolves most of my questions especially those for Section 4. I do think that the work should emphasize more on Transformer side (which is exactly the main focus of this paper) and tone down its application to hypergraphs. Still, I think the new experiments comparing with Hyper-SAGNN and S2G are great to be included in the revision. I will raise my score to 6.

**Time Spent Reviewing:**

8

---

> ### Author Response · Authors · 2021-08-10
> **Response to R4 (tk1s) - 1/3**
>
> We thank the reviewer for the detailed comments. Below we address the main concerns.
>
> ---------------------------------------------------------------------------------------------------------------------------
>
> Q1-1. “In Section 4.1, line 191-198. In the example of $L_{1\to1}$, the authors mention that they only use the basis $I_n$ and ignore the sum-pooling $1_n1_n^\top$ in Eq. (4). Then they claim this light-weighted design is still able to model “interactions” between input elements. However, if we drop the sum-pooling $1_n1_n^\top$ in Eq. (4), then no interactions between elements are considered, as now it only uses the basis $I_n$. This part really confuses me. Please elaborate more.”
>
> A1-1. Let us clarify; the lightweight layers themselves contain no interactions, but are used within $\text{Enc}{k\to l}$ to generate query and key projections of the input (see line 194). Then, the element dependency within $\text{Enc}_{k\to l}$ is handled by the higher-order self-attention that faithfully uses all equivalence classes, as in Eq. (12). Please note that this design choice is coherent with the original (first-order) Transformers, where the elements are first linearly transformed to query and key using *element-wise* basis ($I_n$) and interactions across the elements are handled by attention (implicitly $1_n1_n^\top$).
>
> Q1-2. “Note that in the simple graph example, the proposed light-weighted design of $L_{1\to1}$ is to ignore all “edge features” which means the graph information is completely neglected. Clearly, it is impossible to recover this information barely from attention among node features in general.”
>
> A1-2. We think that our illustration in line 192-198 could be a bit confusing and want to clarify it. The first-order linear layer $L_{1\to1}$ is used for set-to-set mapping (a set is a first-order tensor) without any edges. To process a graph (with edges), one needs to use second-order layers such as $L_{2\to2}$, which does not ignore edge information. Just to be sure, the light-weighted design of $L_{2\to2}$ does not ignore edge features as well.
>
> ---------------------------------------------------------------------------------------------------------------------------
>
> Q2. “The heuristic approach of Section 4.2, especially the use of $E’$ is unclear. Since the authors say it is a heuristic, I expect that this approach will lose some information in (at least) some special cases? Please provide a minimal illustrative example (i.e. $k=2, l=1$?) to explain the use of $E’$.”
>
> A2. Formally, the procedure of converting $E$ to $E’$ is network projection (see the illustration in Fig 1 of [Carletti et al., 2020](https://www.researchgate.net/publication/339341513_Random_walks_on_hypergraphs)) generalized to any input and output orders. Some illustrative examples of the projection are as follows: $k=2 \to l=1$ gives us the set of individual nodes in the graph, $k=3 \to l=2$ gives us edges from sides of triangles (mesh), and generally, $k \to l=2$ gives us a clique for each order-$k$ hyperedge. We chose this heuristic for order conversion because network projection is an often-used reasonable technique for hypergraph reduction.
>
> ---------------------------------------------------------------------------------------------------------------------------
>
> Q3. “Line 240. The definition of set I is unclear. Does it mean the set that union all $\mathbf{i}$ for all $(\mathbf{i}, \mathbf{j})\in\mu$ over all possible $\mathbf{j}$? If this is the case, then shouldn’t the term that we actually summing over in Eq. (19) “less” than Eq. (20)? Why this resulting in further reduction needs more elaboration.”
>
> A3. As the reviewer pointed out, we decouple query index $\mathbf{j}$ and respective key summation scope  $\{\mathbf{i}|(\mathbf{i}, \mathbf{j})\in\mu\}$ in Eq. (20) by simply taking union over all  $\mathbf{j}$, so $\mathcal{I} = \cup_\mathbf{j}\{\mathbf{i}|(\mathbf{i}, \mathbf{j})\in\mu\}$. Let us explain why it reduces computation when we use kernel attention. We will explain the first-order case ($\text{Enc}_{1\to1}$) for illustration purposes, but the argument readily extends to all orders.
>
> In $\text{Enc}{1\to1}$, given $n$ input entries, let us introduce kernel attention and write $o_j = \sum_{\{i | i \neq j\}}(q_j^\top k_i) v_i$ (feature map $\phi$ and normalization omitted for brevity). Collectively, this requires $n(n-1)$ summations because for each $j$-th query we need to sum over all $i \neq j$. By decoupling we write $o_j = \sum_i(q_j^\top k_i) v_i$. Here, because decoupling removes dependency between query index and key indices to sum ($\mathcal{I} = \{1, …, n\}$), we can take the query out of the summation and rewrite $o_j = q_j^\top\sum_i k_i v_i$. This allows us to first compute $\sum_i k_i v_i$ once ($n$ summations), save it in memory, and reuse it for all queries without additional summation (this is a general trick employed in kernel attention methods ([Katharopoulos et al., 2020](https://arxiv.org/pdf/2006.16236.pdf), [Choromanski et al., 2021](https://arxiv.org/pdf/2009.14794.pdf))). Consequently, $n(n-1)$ summations have reduced to $n$ summations due to decoupling.

---

> > ### Author Response · Authors · 2021-08-10
> > **Response to R4 (tk1s) - 2/3**
> >
> > ---------------------------------------------------------------------------------------------------------------------------
> >
> > Q4. “On the experiment side, since I feel the main contribution of the current paper is reducing the complexity of higher-order Transformer, I think the authors should at least conduct one experiment on it. More precisely, for a fixed hypergraph with m hyperedges, I expect to see the experiment showing the time complexity of the proposed model remains unchanged even for increasing order $k$, $l$ in $\text{Enc}_{k\to l}$. In line 327-334. The authors mention that due to the growth of $k$th-order bell number, the higher-order model is actually infeasible. In this case, I feel the importance and contribution of the current paper are greatly degraded.
> >
> > A4. Let us clarify that our asymptotic analysis is with respect to the increasing size of input hypergraph ($n$ or $m$), and we consider the orders $k$ and $l$ as fixed constants as they do not change once we fix a model. If we wish to increase the order of data, the model has to be redesigned with an increased number of equivalence classes following $\text{bell}(k+l)$. This is a common challenge of all tensor-based graph neural networks, including ours, [Maron et al., 2018](https://arxiv.org/pdf/1812.09902.pdf), [Maron et al., 2019a](http://proceedings.mlr.press/v97/maron19a/maron19a.pdf), [Keriven et al., 2019](https://proceedings.neurips.cc/paper/2019/file/ea9268cb43f55d1d12380fb6ea5bf572-Paper.pdf), and [Serviansky et al., 2020](https://arxiv.org/pdf/2002.08772.pdf). We will revise the paper and tone down the claims on hypergraphs so that this is more clear to the readers. Please note that a related work by [Maron et al., 2019b](https://arxiv.org/pdf/1905.11136.pdf) also formulated general order-$k$ GNNs, but they analyzed asymptotic complexity in the fixed, second order only. Still, their model is $O(n^2)$ and is already worse than ours in the second order ($O(m)$).
> >
> > Also, please consider that our contribution of reducing the asymptotic complexity from $O(n^{k+l})$ to $O(m)$ given fixed $k$ and $l$ is significant. This is because in lower-orders, the polynomial $n^{k+l}$ is the bottleneck that prevents the adaptation of permutation equivariant models such as of [Maron et al., 2018](https://arxiv.org/pdf/1812.09902.pdf). For instance, for $k=l=2$ (graphs), $\text{Enc}{2\to2}$ and $L_{2\to2}$ are both $O(n^4)$ which is prohibitive; our approach reduces the complexity to $O(m)$, which is very practical while still more expressive than all message-passing networks as we demonstrated in the paper.
> >
> > To experimentally verify our claims on linear complexity in fixed-order, we conducted a runtime and memory consumption analysis for 4-layer second-order models using Barabási-Albert random graphs with varying numbers of nodes and edges. The results are summarized below ($n$: number of nodes, $m$: number of edges).
> >
> > Barabási-Albert (# edges for preferential attachment = 5)
> >
> > | $n$                                | 10    | 50    | 100    | 1k    | 2k    | 4k    | 6k    | 8k    | 10k   | 20k    |
> > |------------------------------------|-------|-------|--------|-------|-------|-------|-------|-------|-------|--------|
> > | $m$                                | 50    | 250   | 500    | 5k    | 10k   | 20k   | 30k   | 40k   | 50k   | 100k   |
> > | Runtime (s)                        |       |       |        |       |       |       |       |       |       |        |
> > | $L_{2\to2}$                        | 0.019 | 0.289 | 2.101  | OOM   | OOM   | OOM   | OOM   | OOM   | OOM   | OOM    |
> > | $\text{Enc}_{2\to2}$               | 0.077 | 0.162 | 1.656  | OOM   | OOM   | OOM   | OOM   | OOM   | OOM   | OOM    |
> > | $\text{Enc}_{2\to2,\phi}$          | 0.123 | 0.142 | 0.171  | OOM   | OOM   | OOM   | OOM   | OOM   | OOM   | OOM    |
> > | $L_{2\to2,\text{S}}$               | 0.048 | 0.098 | 0.161  | OOM   | OOM   | OOM   | OOM   | OOM   | OOM   | OOM    |
> > | $\text{Enc}_{2\to2,\text{S}}$      | 0.117 | 0.145 | 0.185  | OOM   | OOM   | OOM   | OOM   | OOM   | OOM   | OOM    |
> > | $\text{Enc}_{2\to2,\text{S},\phi}$ | 0.148 | 0.152 | 0.135  | 0.227 | 0.235 | 0.368 | 0.420 | 0.492 | 0.608 | 0.910  |
> > | Peak memory (GB)                   |       |       |        |       |       |       |       |       |       |        |
> > | $L_{2\to2}$                        | 0.002 | 0.159 | 2.077  | OOM   | OOM   | OOM   | OOM   | OOM   | OOM   | OOM    |
> > | $\text{Enc}_{2\to2}$               | 0.005 | 1.299 | 19.584 | OOM   | OOM   | OOM   | OOM   | OOM   | OOM   | OOM    |
> > | $\text{Enc}_{2\to2,\phi}$          | 0.004 | 0.074 | 0.295  | OOM   | OOM   | OOM   | OOM   | OOM   | OOM   | OOM    |
> > | $L_{2\to2,\text{S}}$               | 0.002 | 0.048 | 0.197  | OOM   | OOM   | OOM   | OOM   | OOM   | OOM   | OOM    |
> > | $\text{Enc}_{2\to2,\text{S}}$      | 0.117 | 0.145 | 0.185  | OOM   | OOM   | OOM   | OOM   | OOM   | OOM   | OOM    |
> > | $\text{Enc}_{2\to2,\text{S},\phi}$ | 0.005 | 0.041 | 0.085  | 0.904 | 1.814 | 3.576 | 5.355 | 7.157 | 8.946 | 17.781 |
> >
> > We omitted error bars due to space constraints (variance was generally low).
> >
> > Consistent with the claims in our paper, we observe that the computation complexity of sparse kernel $\text{Enc}_{2\to2}$ linearly scales to the number of edges in terms of memory, and achieves near-linear time similar to Performer ([Choromanski et al., 2021](https://arxiv.org/pdf/2009.14794.pdf)) that leverages kernel attention. Also, it is the only variant that successfully and consistently scales to graphs with 100k edges, while still very fast in sparser or smaller graphs.
> > We appreciate the comment and will add this empirical analysis to the paper.
> >
> > Additionally, the $\text{bell}(k+l)$ number of equivalence classes in higher-orders can be reduced e.g., by assuming a certain structure of input and output. Let us provide a quick example. For $k$-uniform hyperedge prediction tasks (e.g., user-location-activity relation prediction from feature set), the number of equivalence classes ($\text{bell}(1+k)$) can be reduced to a practical level ($1+k$) without any decrease in the model expressibility. We leverage this property in the hyperedge prediction experiments that we introduce in A5. Formally, we show the following:
> >
> > Proposition. If input is first-order (node set) and we only consider output hyperedges without loops ($\mathbf{j}$-th output entry where $\mathbf{j}$1, ..., $\mathbf{j}$k are unique), $L_{1\to k}$ or $\text{Enc}_{1\to k}$ can be implemented using only ($1+k$) equivalence classes ($\mu$‘s) instead of $\text{bell}(1+k)$.
> >
> > Proof. As we only care about output hyperedges with unique index elements, only $\mu$‘s that correspond to partitions of $[k+1]$ with entries $[k]$ contained in distinct subsets contribute to output. There are $(k+1)$ such partitions depending on where the last entry $(k+1)$ belongs to, so it is sufficient that we have $(k+1)$ $\mu$.
> >
> > We believe that reducing $\text{bell}(k+l)$ equivalence classes in general cases is an important next research direction due to the high flexibility of our framework (handling graphs with any order, learning graph mappings across different orders, etc.) that is not readily available for the current graph neural networks.

---

> > > ### Author Response · Authors · 2021-08-10
> > > **Response to R4 (tk1s) - 3/3**
> > >
> > >
> > > ---------------------------------------------------------------------------------------------------------------------------
> > >
> > > Q5. “Since the paper mainly focuses on the case where the input is a hypergraph, I think the authors should have some experiments regarding hypergraphs. However, neither any experiments on hypergraphs nor hypergraph neural network baselines are included in the experiment section. I suggest the authors either conduct experiments on hypergraphs and compare with SOTA hypergraph neural networks, or tone down the emphasis of hypergraphs at least in the abstract and introduction.”
> > >
> > > A5. We appreciate the reviewer for the valuable comment. As suggested by the reviewer, we implemented and tested our model for k-uniform hypergraph prediction task (e.g., user-location-activity prediction), and compared with S2G (Section 5) and the state-of-the-art, self-attention based Hyper-SAGNN ([Zhang et al., 2020](https://openreview.net/pdf?id=ryeHuJBtPH)) in 3 datasets for the transductive 3-edge prediction task. The results are outlined below. We reproduced the scores for Hyper-SAGNN from the authors’ code and took the scores for other baselines from [Zhang et al., 2020](https://openreview.net/pdf?id=ryeHuJBtPH).
> > >
> > > |                                     | GPS   |       |                      | MovieLens |       |                      | Drug  |       |
> > > |-------------------------------------|-------|-------|----------------------|-----------|-------|----------------------|-------|-------|
> > > |                                     | AUC   | AUPR  |                      | AUC       | AUPR  |                      | AUC   | AUPR  |
> > > | node2vec-mean ([Grover et al., 2016](https://arxiv.org/pdf/1607.00653.pdf)) | 0.563 | 0.191 |                      | 0.562     | 0.197 |                      | 0.67  | 0.246 |
> > > | node2vec-min ([Grover et al., 2016](https://arxiv.org/pdf/1607.00653.pdf))  | 0.57  | 0.185 |                      | 0.539     | 0.186 |                      | 0.684 | 0.258 |
> > > | DHNE ([Tu et al., 2018](https://arxiv.org/pdf/1711.10146.pdf))              | 0.91  | 0.668 |                      | 0.877     | 0.668 |                      | 0.925 | 0.859 |
> > > | Hyper-SAGNN-E ([Zhang et al., 2020](https://openreview.net/pdf?id=ryeHuJBtPH))  | 0.947 | 0.788 |                      | 0.922     | **0.792** |                      | 0.963 | 0.897 |
> > > | Hyper-SAGNN-W ([Zhang et al., 2020](https://openreview.net/pdf?id=ryeHuJBtPH))  | 0.907 | 0.632 |                      | 0.909     | 0.683 |                      | 0.956 | 0.89  |
> > > | S2G                                 | 0.943 | 0.726 |                      | 0.918     | 0.737 |                      | 0.963 | 0.898 |
> > > | Ours ($\phi$)                            | **0.952** | **0.804** |                      | **0.923**     | 0.771 |                      | **0.964** | **0.901** |
> > >
> > > Our model outperforms S2G in all datasets, and Hyper-SAGNN in most cases (all but one metric in MovieLens dataset). The results suggest that our proposed framework, specifically higher-order self-attention, is effective in learning higher-order representation beyond second-order graphs. Note that we did not introduce any form of task-specific heuristics into the model architecture, while some of the compared methods such as Hyper-SAGNN depend on many inductive biases (e.g., separated static/dynamic branches, Hadamard power, etc.).

---

> > > > ### Comment · Reviewer_tk1s · 2021-08-20
> > > > **Thanks for the detailed response!**
> > > >
> > > > I really appreciate the detailed response by the authors, which resolves almost all my questions especially those for Section 4. I do think that the work should emphasize more on Transformer side (which is exactly the main focus of this paper) and tone down its application to hypergraphs. Still, I think the new experiments comparing with Hyper-SAGNN and S2G are great to be included in the revision. I will raise my score to 6.

---

### Official Review · Reviewer_uPAM · 2021-07-15

**Rating:** 6
**Confidence:** 3

**Summary:**

In this paper, the authors proposed a transformer layer that generalizes the equivariant linear layers with self-attention to higher order tensor input. They defined the attention coefficient for each equivalence class $\mu$ with indices satisfied $(i, j) \in \mu$ to reduce unnecessary computation between queries and keys matrices. To further reduce the computation complexity of higher order transformers, the authors provides 3 steps of approximation: i) a lightweight linear layers defined only for subset of equivalence class to decouple the indices $i$ and $j$. ii) adding proxy hyperedges $E'$ that is contained in original hyperedges $E$ and help reducing the computation to only $m$ hyperedges across different layers. iii) a kernel trick to decompose the attention coefficients into kernel map of query and keyword matrices.

**Limitations And Societal Impact:**

The authors addressed the limitation in Section 6 and potential social impact in the supplementary.

**Main Review:**

pros:
- Generalized higher order linear layers with transformer encoder with self attention.
- Reduced the complexity of transformer through above-mentioned approximations.
- Overall, the transformer does have a larger expressive power than simple first order message passing algorithms.
- Extensive and detailed experiments with synthetic and real world datasets in different graph-related tasks.

cons:
- I am concerned about sharing queries $\tilde Q$ and keys $\tilde K$ across all equivalence classes, as they might correspond to completely different propagation schemes with various bases.
- Most of the results are still order 1 or 2 transformer layers, and the dataset are ordinary graphs. Lack of actual higher order interactions and hypergraph applications.
- Some idea of proof is preferable in the main text instead of leaving all to supplementary.
- The complexity is regarding to the number of matrix multiplication, but it can still be time/memory consuming in cases of large graph with higher feature dimension. An running time comparison in the experiment section would be a nice add-on for showcasing the efficiency.
- The approximation of kernel attention in equation (20) removes correlations between $j$ and $i$ and shared across different input indices $i$ seems like a big trade-off and may raise some questions on whether the attention mechanism is still needed.
- More detail on how the edge/node features of synthetic data are generated, how train/valid/test split are done with the chained labeled data is also missing.
- Dataset statistics are missing.
- Table 1 does not have any standard error.



**Time Spent Reviewing:**

10

---

> ### Author Response · Authors · 2021-08-10
> **Response to R3 (uPAM) - 1/2**
>
>
> Response to R3 (uPAM)
> We thank the reviewer for the detailed comments. Below we address the main concerns.
>
> ---------------------------------------------------------------------------------------------------------------------------
>
> Q1. “I am concerned about sharing queries and keys across all equivalence classes, as they might correspond to completely different propagation schemes with various bases.”
>
> A1. We appreciate the reviewer for the valuable comment. Weight sharing is the implementation trick we initially used to reduce memory/parameters, but it may limit the modeling capacity as suggested by the reviewer. We found that a simpler yet better alternative is to have separate low-dimensional queries and keys without sharing. In our ablation study of 3-folds experiment on the IMDB-M dataset, we observed that having separate low-dimensional queries and keys indeed improves the performance. More interestingly, visualization of the learned attention also showed that the unsharing allows the model to learn completely different dependencies (local/global) depending on equivalence class even for the same Q/K order. We appreciate the comment and will update the results of the main paper accordingly. For the experiments on OGB and hyperedge prediction added in the rebuttal, we unshared the Q/K.
>
> |                             | Validation acc. (%) |
> |-----------------------------|---------------------|
> | Ours ($\phi$), Q/K shared   | 52.44 ± 3.14        |
> | Ours ($\phi$), Q/K unshared | **54.0 ± 3.81**     |
>
> ---------------------------------------------------------------------------------------------------------------------------
>
> Q2. “Most of the results are still of order 1 or 2 transformer layers, and the dataset are ordinary graphs. Lack of actual higher order interactions and hypergraph applications.”
>
> A2. We appreciate the reviewer for the valuable comment. As suggested by the reviewer, we implemented and tested our model for $k$-uniform hypergraph prediction task (e.g., user-location-activity prediction), and compared with S2G (Section 5) and the state-of-the-art, self-attention based Hyper-SAGNN ([Zhang et al., 2020](https://openreview.net/pdf?id=ryeHuJBtPH)) in 3 datasets for the transductive 3-edge prediction task. The results are outlined below. We reproduced the scores for Hyper-SAGNN from the authors’ code and took the scores for other baselines from [Zhang et al., 2020](https://openreview.net/pdf?id=ryeHuJBtPH).
>
> |                                     | GPS   |       |                      | MovieLens |       |                      | Drug  |       |
> |-------------------------------------|-------|-------|----------------------|-----------|-------|----------------------|-------|-------|
> |                                     | AUC   | AUPR  |                      | AUC       | AUPR  |                      | AUC   | AUPR  |
> | node2vec-mean ([Grover et al., 2016](https://arxiv.org/pdf/1607.00653.pdf)) | 0.563 | 0.191 |                      | 0.562     | 0.197 |                      | 0.67  | 0.246 |
> | node2vec-min ([Grover et al., 2016](https://arxiv.org/pdf/1607.00653.pdf))  | 0.57  | 0.185 |                      | 0.539     | 0.186 |                      | 0.684 | 0.258 |
> | DHNE ([Tu et al., 2018](https://arxiv.org/pdf/1711.10146.pdf))              | 0.91  | 0.668 |                      | 0.877     | 0.668 |                      | 0.925 | 0.859 |
> | Hyper-SAGNN-E ([Zhang et al., 2020](https://openreview.net/pdf?id=ryeHuJBtPH))  | 0.947 | 0.788 |                      | 0.922     | **0.792** |                      | 0.963 | 0.897 |
> | Hyper-SAGNN-W ([Zhang et al., 2020](https://openreview.net/pdf?id=ryeHuJBtPH))  | 0.907 | 0.632 |                      | 0.909     | 0.683 |                      | 0.956 | 0.89  |
> | S2G                                 | 0.943 | 0.726 |                      | 0.918     | 0.737 |                      | 0.963 | 0.898 |
> | Ours ($\phi$)                            | **0.952** | **0.804** |                      | **0.923**     | 0.771 |                      | **0.964** | **0.901** |
>
> Our model outperforms S2G in all datasets, and Hyper-SAGNN in most cases (all but one metric in MovieLens dataset). The results suggest that our proposed framework, specifically higher-order self-attention, is effective in learning higher-order representation beyond second-order graphs. Note that we did not introduce any form of task-specific heuristics into the model architecture, while some of the compared methods such as Hyper-SAGNN depend on many inductive biases (e.g., separated static/dynamic branches, Hadamard power, etc.).
>
> ---------------------------------------------------------------------------------------------------------------------------
>
> Q3. “Some idea of proof is preferable in the main text instead of leaving it all to supplementary.”
>
> A3. We appreciate the comment and will move the sketches of proofs for Proposition 1 and Theorem 2 to the main text.
>
> ---------------------------------------------------------------------------------------------------------------------------
>
> Q4. “The complexity is regarding the number of matrix multiplication, but it can still be time/memory consuming in cases of large graphs with higher feature dimension. A running time comparison in the experiment section would be a nice add-on for showcasing the efficiency.”
>
> A4. To address this issue, we conducted a runtime and memory consumption analysis for 4-layer second-order models using Barabási-Albert random graphs with varying numbers of nodes and edges. The results are summarized below ($n$: number of nodes, $m$: number of edges).
>
> Barabási-Albert (# edges for preferential attachment = 5)
>
> | $n$                                | 10    | 50    | 100    | 1k    | 2k    | 4k    | 6k    | 8k    | 10k   | 20k    |
> |------------------------------------|-------|-------|--------|-------|-------|-------|-------|-------|-------|--------|
> | $m$                                | 50    | 250   | 500    | 5k    | 10k   | 20k   | 30k   | 40k   | 50k   | 100k   |
> | Runtime (s)                        |       |       |        |       |       |       |       |       |       |        |
> | $L_{2\to2}$                        | 0.019 | 0.289 | 2.101  | OOM   | OOM   | OOM   | OOM   | OOM   | OOM   | OOM    |
> | $\text{Enc}_{2\to2}$               | 0.077 | 0.162 | 1.656  | OOM   | OOM   | OOM   | OOM   | OOM   | OOM   | OOM    |
> | $\text{Enc}_{2\to2,\phi}$          | 0.123 | 0.142 | 0.171  | OOM   | OOM   | OOM   | OOM   | OOM   | OOM   | OOM    |
> | $L_{2\to2,\text{S}}$               | 0.048 | 0.098 | 0.161  | OOM   | OOM   | OOM   | OOM   | OOM   | OOM   | OOM    |
> | $\text{Enc}_{2\to2,\text{S}}$      | 0.117 | 0.145 | 0.185  | OOM   | OOM   | OOM   | OOM   | OOM   | OOM   | OOM    |
> | $\text{Enc}_{2\to2,\text{S},\phi}$ | 0.148 | 0.152 | 0.135  | 0.227 | 0.235 | 0.368 | 0.420 | 0.492 | 0.608 | 0.910  |
> | Peak memory (GB)                   |       |       |        |       |       |       |       |       |       |        |
> | $L_{2\to2}$                        | 0.002 | 0.159 | 2.077  | OOM   | OOM   | OOM   | OOM   | OOM   | OOM   | OOM    |
> | $\text{Enc}_{2\to2}$               | 0.005 | 1.299 | 19.584 | OOM   | OOM   | OOM   | OOM   | OOM   | OOM   | OOM    |
> | $\text{Enc}_{2\to2,\phi}$          | 0.004 | 0.074 | 0.295  | OOM   | OOM   | OOM   | OOM   | OOM   | OOM   | OOM    |
> | $L_{2\to2,\text{S}}$               | 0.002 | 0.048 | 0.197  | OOM   | OOM   | OOM   | OOM   | OOM   | OOM   | OOM    |
> | $\text{Enc}_{2\to2,\text{S}}$      | 0.117 | 0.145 | 0.185  | OOM   | OOM   | OOM   | OOM   | OOM   | OOM   | OOM    |
> | $\text{Enc}_{2\to2,\text{S},\phi}$ | 0.005 | 0.041 | 0.085  | 0.904 | 1.814 | 3.576 | 5.355 | 7.157 | 8.946 | 17.781 |
>
> We omitted error bars due to space constraints (variance was generally low).
>
> Consistent with the claims in our paper, we observe that the computation complexity of sparse kernel $\text{Enc}_{2\to2}$ linearly scales to the number of edges in terms of memory, and achieves near-linear time similar to Performer ([Choromanski et al., 2021](https://arxiv.org/pdf/2009.14794.pdf)) that leverages kernel attention. Also, it is the only variant that successfully and consistently scales to graphs with 100k edges, while still very fast in sparser or smaller graphs.
> We appreciate the comment and will add this empirical analysis to the paper.

---

> > ### Author Response · Authors · 2021-08-10
> > **Response to R3 (uPAM) - 2/2**
> >
> >
> > ---------------------------------------------------------------------------------------------------------------------------
> >
> > Q5. “The approximation of Eq. (20) that removes correlation between $\mathbf{j}$ and $\mathbf{i}$ seems like a big trade-off and may raise some questions on whether the attention mechanism is still needed.”
> >
> > A5. We would like to clarify that the decoupling of summation of key-value products ($\mathbf{i}$) and query-wise application ($\mathbf{j}$) is a general trick employed by kernel attention methods ([Katharopoulos et al., 2020](https://arxiv.org/pdf/2006.16236.pdf), [Choromanski et al., 2021](https://arxiv.org/pdf/2009.14794.pdf)). Although the decoupling of $\mathbf{i}$ and $\mathbf{j}$ during computation might seem destructive, the method is theoretically and empirically guaranteed, and a suitable choice of kernel can even accurately approximate or outperform softmax attention ([Choromanski et al., 2021](https://arxiv.org/pdf/2009.14794.pdf)). Our approximation of taking union $\cup_\mathbf{j}\{\mathbf{i}|(\mathbf{i},\mathbf{j})\in\mu\}$
> > additionally allows a query to attend to some more keys (depending on $\mu$), but we emphasize that it does not hurt the central requirement of attention that each query can assign different attention weights to the keys.
> >
> > ---------------------------------------------------------------------------------------------------------------------------
> >
> > Q6. “More detail on how the edge/node features of synthetic data are generated, how train/valid/test split is done with the chained labeled data is also missing.”
> >
> > A6. We provided the detailed descriptions in the supplementary file (section A.3.1.) due to space limitations but will move it to the main text. For the synthetic chains, we used a small dataset composed of 40 training chains each with 20 nodes, and 20 test chains each with 200 nodes. Each chain is randomly assigned with a binary label, which is encoded as a one-hot vector only at the terminal node. The goal is to classify all nodes in the chain according to the label. We trained the model for a fixed number of epochs (100 epochs) without validation data.
> >
> > ---------------------------------------------------------------------------------------------------------------------------
> >
> > Q7. “Dataset statistics are missing.”
> >
> > A7. We appreciate the comment. We provide the dataset statistics below. We will update the paper accordingly.
> >
> > | Dataset     | MUTAG | PTC  | PROTEINS | COLLAB | IMDB-B | IMDB-M |
> > |-------------|-------|------|----------|--------|--------|--------|
> > | Size        | 188   | 344  | 1113     | 5000   | 1000   | 1500   |
> > | # classes   | 2     | 2    | 2        | 3      | 2      | 3      |
> > | Avg. # node | 17.9  | 14.3 | 39.1     | 74.5   | 19.7   | 13     |
> > | Avg. # edge | 19.8  | 14.7 | 72.8     | 2457.8 | 96.5   | 65.9   |
> >
> > | Dataset     | Chains                  |
> > |-------------|-------------------------|
> > | Size        | 60                      |
> > | # classes   | 2                       |
> > | Avg. # node | 20 (train) / 200 (test) |
> >
> > | Dataset     | Jets | Delaunay (50) | Delaunay (20-80) |
> > |-------------|------|---------------|------------------|
> > | Size        | 0.9M | 55k           | 55k              |
> > | Avg. # node | 7.11 | 50            | 50               |
> > | Avg. # edge | 35.9 | 273.6         | 273.9            |
> >
> > | Dataset    | GPS                      | MovieLens        | Drug                 |
> > |------------|--------------------------|------------------|----------------------|
> > | Node types | user, location, activity | user, movie, tag | user, drug, reaction |
> > | # node     | 146, 70, 5               | 2113, 5908, 9079 | 12, 1076, 6398       |
> > | # edge     | 1436                     | 47957            | 171756               |
> >
> > | Dataset     | PCQM4M-LSC |
> > |-------------|------------|
> > | Size        | 3.8M       |
> > | Avg. # node | 14.1       |
> > | Avg. # edge | 14.6       |
> >
> > ---------------------------------------------------------------------------------------------------------------------------
> >
> > Q8. “Table 1 does not have any standard error.”
> >
> > A8. We appreciate the comment. Below we report the average and standard deviation of 10 fold experiments. We observe that our method is the only one that achieves 100% accuracy in this task. We will update the table accordingly.
> >
> > | Method                      | Micro-F1 (%) | Macro-F1 (%) |
> > |-----------------------------|--------------|--------------|
> > | GCN                         | 47.78 ± 4.17 | 33.58 ± 1.86 |
> > | GIN-0                       | 53.72 ± 4.17 | 36.22 ± 1.86 |
> > | GAT                         | 47.78 ± 4.17 | 33.58 ± 1.86 |
> > | MLP2 (S)                    | 53.5 ± 4.16  | 36.04 ± 1.97 |
> > | Ours (S) w/o global         | 53.72 ± 4.17 | 36.22 ± 1.86 |
> > | Ours (S, $\phi$) w/o global | 50.77 ± 5.15 | 35.22 ± 2.17 |
> > | Ours (S)                    | **100 ± 0**  | **100 ± 0**  |
> > | Ours (S, $\phi$)            | **100 ± 0**  | **100 ± 0**  |

---

> > > ### Comment · Reviewer_uPAM · 2021-08-22
> > > **Thanks for the detail response**
> > >
> > > Thank you for answering most of my questions and I am more confident to raise my score to 6. Please add the results of separate low-dimensional queries and keys without sharing to the revision, as it showcases more the expressiveness of the model.

---

### Official Review · Reviewer_fYvj · 2021-07-16

**Rating:** 7
**Confidence:** 4

**Summary:**

This paper generalizes Transformer to higher-order Transformer, which is higher-order permutation invariant. To reduce the complexity, the authors propose to use sparse higher-order Transformer to achieve quadratic complexity, and further adopt the kernel attention approach to linear complexity.

**Ethical Concerns:**

No.

**Limitations And Societal Impact:**

Yes.

**Main Review:**

Pros:
- This paper is well-written and well-organized, which is easy to follow.
- The idea taht extend the concept high-order permutation invirant to Transformer, or say self-attention, is novel and indeed practical.
- The design of higher-order self-attention makes sense to me.
- The propsed model works well on synthetic dataset, which demonstrates that the model could match the motivation of this paper.

Cons:
- It lacks a theoretical analysis of the performance (expressivenss, convergence, etc) about the two efficient tricks: sparse and kernel trick.
- It lacks a empirical study about the running time in terms of the number of the nodes.
- The original paper efficient transformers with kernel tricks (Performer, Transformers are RNNs, etc)  that cliam the performance would be degenerated if the models are trained from scratch. I wonder how the proposed higher-order Transformer with kernel trick solve this issue?
- The baseline methods are too weak and old, most of the baseline methods are published at or before 2019. Even though, the proposed method could not outperform all baseline methods on all datasets.

One possible reason is that, the datasets used in this paper are relative small, and Transformer would be benefited from massive data (see Bert or ViT). Please consider to use large dataset to demonstrate your method.





**Time Spent Reviewing:**

5

---

> ### Author Response · Authors · 2021-08-10
> **Response to R2 (fYvj)**
>
> We thank the reviewer for the detailed comments. Below we address the main concerns.
>
> ---------------------------------------------------------------------------------------------------------------------------
>
> Q1. “It lacks a theoretical analysis of the performance (expressiveness, convergence, etc) about the two efficient tricks: sparse and kernel trick.”
>
> A1. We appreciate the comment. However, please note that Section 4.4 provides a theoretical analysis of the expressiveness of our model with two efficient tricks. Specifically, we proved in Theorem 2 that the two-layer composition of our encoder layers *with* sparsity and kernel attention can approximate any message-passing neural networks within the framework of ([Gilmer et al., 2017](http://proceedings.mlr.press/v70/gilmer17a/gilmer17a.pdf)) with arbitrary precision. This leads to a corollary (Corollary 1) that our method *with* sparsity and kernel attention can be strictly more expressive than any message passing neural networks. We will revise the title of Section 4.4 to make this proof more visible to readers. We are also happy to elaborate more during the rolling-discussion period if the reviewer has any remaining concerns.
>
> ---------------------------------------------------------------------------------------------------------------------------
>
> Q2. “It lacks an empirical study about the running time in terms of the number of the nodes.”
>
> A2. To address this issue, we conducted a runtime and memory consumption analysis for 4-layer second-order models using Barabási-Albert random graphs with varying numbers of nodes and edges. The results are summarized below ($n$: number of nodes, $m$: number of edges).
>
> Barabási-Albert (# edges for preferential attachment = 5)
>
> | $n$                                | 10    | 50    | 100    | 1k    | 2k    | 4k    | 6k    | 8k    | 10k   | 20k    |
> |------------------------------------|-------|-------|--------|-------|-------|-------|-------|-------|-------|--------|
> | $m$                                | 50    | 250   | 500    | 5k    | 10k   | 20k   | 30k   | 40k   | 50k   | 100k   |
> | Runtime (s)                        |       |       |        |       |       |       |       |       |       |        |
> | $L_{2\to2}$                        | 0.019 | 0.289 | 2.101  | OOM   | OOM   | OOM   | OOM   | OOM   | OOM   | OOM    |
> | $\text{Enc}_{2\to2}$               | 0.077 | 0.162 | 1.656  | OOM   | OOM   | OOM   | OOM   | OOM   | OOM   | OOM    |
> | $\text{Enc}_{2\to2,\phi}$          | 0.123 | 0.142 | 0.171  | OOM   | OOM   | OOM   | OOM   | OOM   | OOM   | OOM    |
> | $L_{2\to2,\text{S}}$               | 0.048 | 0.098 | 0.161  | OOM   | OOM   | OOM   | OOM   | OOM   | OOM   | OOM    |
> | $\text{Enc}_{2\to2,\text{S}}$      | 0.117 | 0.145 | 0.185  | OOM   | OOM   | OOM   | OOM   | OOM   | OOM   | OOM    |
> | $\text{Enc}_{2\to2,\text{S},\phi}$ | 0.148 | 0.152 | 0.135  | 0.227 | 0.235 | 0.368 | 0.420 | 0.492 | 0.608 | 0.910  |
> | Peak memory (GB)                   |       |       |        |       |       |       |       |       |       |        |
> | $L_{2\to2}$                        | 0.002 | 0.159 | 2.077  | OOM   | OOM   | OOM   | OOM   | OOM   | OOM   | OOM    |
> | $\text{Enc}_{2\to2}$               | 0.005 | 1.299 | 19.584 | OOM   | OOM   | OOM   | OOM   | OOM   | OOM   | OOM    |
> | $\text{Enc}_{2\to2,\phi}$          | 0.004 | 0.074 | 0.295  | OOM   | OOM   | OOM   | OOM   | OOM   | OOM   | OOM    |
> | $L_{2\to2,\text{S}}$               | 0.002 | 0.048 | 0.197  | OOM   | OOM   | OOM   | OOM   | OOM   | OOM   | OOM    |
> | $\text{Enc}_{2\to2,\text{S}}$      | 0.117 | 0.145 | 0.185  | OOM   | OOM   | OOM   | OOM   | OOM   | OOM   | OOM    |
> | $\text{Enc}_{2\to2,\text{S},\phi}$ | 0.005 | 0.041 | 0.085  | 0.904 | 1.814 | 3.576 | 5.355 | 7.157 | 8.946 | 17.781 |
>
> We omitted error bars due to space constraints (variance was generally low).
>
> Consistent with the claims in our paper, we observe that the computation complexity of sparse kernel $\text{Enc}_{2\to2}$ linearly scales to the number of edges in terms of memory, and achieves near-linear time similar to Performer ([Choromanski et al., 2021](https://arxiv.org/pdf/2009.14794.pdf)) that leverages kernel attention. Also, it is the only variant that successfully and consistently scales to graphs with 100k edges, while still very fast in sparser or smaller graphs.
> We appreciate the comment and will add this empirical analysis to the paper.
>
> ---------------------------------------------------------------------------------------------------------------------------
>
> Q3. “The original paper efficient transformers with kernel tricks (Performer, Transformers are RNNs, etc) claim the performance would be degenerated if the models are trained from scratch. I wonder how the proposed higher-order Transformer with kernel solves this issue?”
>
> A3. In our experiments (Table 1-3), we overall did not experience a significant performance drop coming from the kernel attention. For instance, please find the results on PTC, IMDB-B in Table 2, and Jets in Table 3 where kernel attention outperforms the softmax counterpart. Also, please note that the exact choice of the kernel is orthogonal to our main contribution and we can alternatively incorporate more advanced kernel attention in the literature. In the large-scale experiments added during the rebuttal (Q4), we used theoretically and empirically guaranteed Performer kernel attention ([Choromanski et al., 2021](https://arxiv.org/pdf/2009.14794.pdf)).
>
> ---------------------------------------------------------------------------------------------------------------------------
>
> Q4. “The baseline methods are too weak and old, most of the baseline methods are published at or before 2019. Even so, the proposed method could not outperform all baseline methods on all datasets. One possible reason is that the datasets used in this paper are relatively small, and Transformer would benefit from massive data (see Bert or ViT).”
>
> A4. We appreciate the comment. We agree that the datasets used for graph classification experiments are generally small (180 to 5k data instances) and may prefer smaller models like GCN. To demonstrate our model on a much larger scale, we conducted a preliminary experiment in the PCQM4M-LSC dataset of Open Graph Benchmark ([Hu et al., 2021](https://arxiv.org/pdf/2103.09430.pdf)), which is one of the largest graph datasets containing 3.8M molecular graphs. Due to the limited rebuttal time frame, we did not apply any model/hyperparameter tuning, and our method is trained less than 10% iterations than all compared methods.
>
> | Method             | Validation MAE |
> |--------------------|----------------|
> | MLP-FINGERPRINT    | 0.2044         |
> | GCN                | 0.1684         |
> | GIN                | 0.1536         |
> | GCN + virtual node | 0.1510         |
> | GIN + virtual node | 0.1396         |
> | Ours (S, $\phi$)        | **0.1376**         |
>
> Although the results are preliminary, we observe promising results; our model surpasses all MPNN baselines, even the ones with a virtual node that can model long-range interactions. The result suggests that using higher-order attention to model long-range dependency is beneficial in large-scale modeling, potentially better than the current practice of augmenting MPNNs with global interaction. Since it is a preliminary result, we expect much better performance if we properly tune the model and train it longer. We appreciate the comment and will add the revised results to the paper.
>
> In addition to experiments on the OGB dataset, please note that the datasets used for set-to-graph experiments in Section 5 contain a fairly large amount of data (0.9M for Jets, 55k for Delaunay a/b), and our method substantially outperforms all baselines in these datasets. It also supports that our design is generic across domains and is effective in the presence of larger data.
>
> Orthogonal to this result, we would like to emphasize that our contribution is building a general equivariant graph neural network that can be universally applied to any tensor-to-tensor mapping. Its application is not only limited to simple graph classification but also extends to much more complex tensor-to-tensor mappings, such as the ones between mixed-order tensors (e.g., set-to-graph prediction in Section 5), which are not directly addressable to the GNNs designed for the graph classification.

---

> > ### Comment · Reviewer_fYvj · 2021-08-11
> > **Thanks for the response**
> >
> > Thanks for addressing my concerns. I'd like to raise my score to 7. Please kindly make sure that corresponding modifications would have been made in the next version.

---

> > ### Comment · Reviewer_fYvj · 2021-08-19
> > **About baseline method**
> >
> > I realize that for PCQM4M-LSC dataset, vanilla Transformer and DeepSet should be the baseline methods. Would you report the performance of those baseline methods?

---

> > > ### Author Response · Authors · 2021-08-19
> > > **More baselines**
> > >
> > > We appreciate the suggestion. We will add the comparison to the suggested baselines as follows:
> > >
> > > - DeepSet for graph (second-order invariant MLP)
> > > - A vanilla Transformer that operates on Laplacian graph node embeddings
> > >
> > > Due to the scale of the PCQM4M-LSC dataset, we expect to get the results sometime in the next week. We will report the updated results as soon as we get one.

---

> > > > ### Comment · Reviewer_fYvj · 2021-08-30
> > > > **More baselines**
> > > >
> > > > I wonder is there any updates about the new baseline results? Hope to see that before the deadline to help me make more objective judegment for this paper.

---

> > > > > ### Author Response · Authors · 2021-08-30
> > > > > **More baselines**
> > > > >
> > > > > We apologize for the delay. Judging from the current training progress, the experiments will be done by 1 Sep (the review deadline is 2 Sep). We will update the results as soon as the experiments are done.

---

> > > > > > ### Author Response · Authors · 2021-09-01
> > > > > > **More baselines**
> > > > > >
> > > > > > As additional baselines, we report performances of a second-order MLP (DeepSet for graphs) and a vanilla (first-order) Transformer in the PCQM4M-LSC dataset of Open Graph Benchmark. As vanilla Transformer operates on node features only, we additionally used Laplacian graph embeddings (Belkin et al., 2003, Dwivedi et al., 2020) as positional embeddings so that the model can consider edge structure information. For all baselines, we set the hidden dimension size, number of layers, and training schedule the same as our second-order Transformer. The results are outlined below.
> > > > > >
> > > > > > | Method | Validation MAE |
> > > > > > |--------------------|----------------|
> > > > > > | MLP$_2$ (S) | 0.1464 |
> > > > > > | Transformer + Laplacian PE | 0.2162 |
> > > > > > | Ours (S, $\phi$) | **0.1376** |
> > > > > >
> > > > > > Perhaps not surprisingly, we observe that both baselines have worse performances compared to the second-order Transformer (our method). The results imply the following: First, comparison to invariant MLP (MLP$_2$ (S)) shows that replacing sum-pooling of invariant MLPs with attention is beneficial in large-scale graph modeling. This observation is consistent with the other experiments in the paper (Table 1,2,3), where the performance of our method improves consistently over the invariant MLP. Second, a comparison to naive Transformer (Transformer + Laplacian PE) shows that augmenting the first-order Transformers with heuristic graph embeddings is insufficient for graph processing. It is mainly because the Laplacian embedding is generally not sufficient to incorporate the features associated with edges, while our method, which can be considered as a second-order Transformer, can naturally utilize all edge information.
> > > > > > In addition to the performance, please note that the computational complexity of the two baselines are much heavier than our method. The asymptotic complexity of (sparse) invariant MLP and the vanilla Transformer are $O(m^2)$ and $O(n^2)$, respectively, while our method has a complexity of $O(m)$.
> > > > > > We appreciate the reviewer’s valuable suggestion and patience. We will add the results together with the OGB experiments to the paper.

---

### Official Review · Reviewer_385N · 2021-07-16

**Rating:** 8
**Confidence:** 4

**Summary:**

The paper proposes a generalization of Transformer architecture to hypergraphs, with a particular focus on developing a fast sparse version.

**Limitations And Societal Impact:**

Limitations paragraph is written greatly and directly calls for subsequent work.

**Main Review:**

The paper is well written and the main motivation is clear. The proposed solution is novel, and, even though not so exciting from the modelling point of view, technically correct. The paper goes at great lengths to improve the scalability of the approach, and appears to do so successfully without compromising on the quality of the solutions and without introducing non-differentiable heuristics.

My main pain point is evaluation these models on the TU graph kernel datasets. As there are around 20-30 datasets, it is easy to overfit the model architecture and design choices to that particular task. While I do not believe this is true, it would be really nice to have some actual higher-order interaction dataset (academic co-authorship), even if it takes some time to produce/process.

Overall, seems like a good paper to appear at the conference.


EDIT: After the authors' comments, I raise my score to 8.

**Time Spent Reviewing:**

4

---

> ### Author Response · Authors · 2021-08-10
> **Response to R1 (385N)**
>
> We thank the reviewer for the positive comments. Below we address the main concerns.
>
> ---------------------------------------------------------------------------------------------------------------------------
>
> Q1. “My main pain point is evaluating these models on the TU graph kernel datasets. As there are around 20-30 datasets, it is easy to overfit the model architecture and design choices to that particular task.”
>
> A1. We appreciate the comment. As noted by the reviewer, TU datasets contain a relatively small number of data instances (180 to 5k). To demonstrate our model on a much larger scale, we conducted a preliminary experiment in the PCQM4M-LSC dataset of Open Graph Benchmark ([Hu et al., 2021](https://arxiv.org/pdf/2103.09430.pdf)), which is one of the largest graph datasets containing 3.8M molecular graphs. Due to the limited rebuttal time frame, we did not apply any model/hyperparameter tuning, and our method is trained less than 10% iterations than all compared methods.
>
> | Method             | Validation MAE |
> |--------------------|----------------|
> | MLP-FINGERPRINT    | 0.2044         |
> | GCN                | 0.1684         |
> | GIN                | 0.1536         |
> | GCN + virtual node | 0.1510         |
> | GIN + virtual node | 0.1396         |
> | Ours (S, $\phi$)        | **0.1376**         |
>
> Although the results are preliminary, we observe promising results; our model surpasses all MPNN baselines, even the ones with a virtual node that can model long-range interactions. The result suggests that using higher-order attention to model long-range dependency is beneficial in large-scale modeling, potentially better than current practice of augmenting MPNNs with global interaction. Since it is a preliminary result, we expect much better performance if we properly tune the model and train it longer. We appreciate the comment and will add the revised results to the paper.
>
> In addition to experiments on the OGB dataset, please note that the datasets used for set-to-graph experiments in Section 5 contain a fairly large amount of data (0.9M for Jets, 55k for Delaunay a/b), and our method substantially outperforms all baselines in these datasets. It also supports that our design is generic across domains and is effective in the presence of larger data.
>
> ---------------------------------------------------------------------------------------------------------------------------
>
> Q2. “It would be really nice to have some actual higher-order interaction dataset (academic co-authorship), even if it takes some time to produce/process.”
>
> A2. We appreciate the reviewer for the valuable comment. As suggested by the reviewer, we implemented and tested our model for k-uniform hypergraph prediction task (e.g., user-location-activity prediction), and compared with S2G (Section 5) and the state-of-the-art, self-attention based Hyper-SAGNN ([Zhang et al., 2020](https://openreview.net/pdf?id=ryeHuJBtPH)) in 3 datasets for the transductive 3-edge prediction task. The results are outlined below. We reproduced the scores for Hyper-SAGNN from the authors’ code and took the scores for other baselines from [Zhang et al., 2020](https://openreview.net/pdf?id=ryeHuJBtPH).
>
> |                                     | GPS   |       |                      | MovieLens |       |                      | Drug  |       |
> |-------------------------------------|-------|-------|----------------------|-----------|-------|----------------------|-------|-------|
> |                                     | AUC   | AUPR  |                      | AUC       | AUPR  |                      | AUC   | AUPR  |
> | node2vec-mean ([Grover et al., 2016](https://arxiv.org/pdf/1607.00653.pdf)) | 0.563 | 0.191 |                      | 0.562     | 0.197 |                      | 0.67  | 0.246 |
> | node2vec-min ([Grover et al., 2016](https://arxiv.org/pdf/1607.00653.pdf))  | 0.57  | 0.185 |                      | 0.539     | 0.186 |                      | 0.684 | 0.258 |
> | DHNE ([Tu et al., 2018](https://arxiv.org/pdf/1711.10146.pdf))              | 0.91  | 0.668 |                      | 0.877     | 0.668 |                      | 0.925 | 0.859 |
> | Hyper-SAGNN-E ([Zhang et al., 2020](https://openreview.net/pdf?id=ryeHuJBtPH))  | 0.947 | 0.788 |                      | 0.922     | **0.792** |                      | 0.963 | 0.897 |
> | Hyper-SAGNN-W ([Zhang et al., 2020](https://openreview.net/pdf?id=ryeHuJBtPH))  | 0.907 | 0.632 |                      | 0.909     | 0.683 |                      | 0.956 | 0.89  |
> | S2G                                 | 0.943 | 0.726 |                      | 0.918     | 0.737 |                      | 0.963 | 0.898 |
> | Ours ($\phi$)                            | **0.952** | **0.804** |                      | **0.923**     | 0.771 |                      | **0.964** | **0.901** |
>
> Our model outperforms S2G in all datasets, and Hyper-SAGNN in most cases (all but one metric in MovieLens dataset). The results suggest that our proposed framework, specifically higher-order self-attention, is effective in learning higher-order representation beyond second-order graphs. Note that we did not introduce any form of task-specific heuristics into the model architecture, while some of the compared methods such as Hyper-SAGNN depend on many inductive biases (e.g., separated static/dynamic branches, Hadamard power, etc.).
>
> Note that our work employs a tensor-based representation of graphs. Although it provides a general representation of the graphs and is theoretically attractive, it is difficult to be directly applied to graphs in varying order (e.g., co-citation network). This is the common challenge in all tensor-based approaches in graph neural networks ([Hartford et al., 2018](http://proceedings.mlr.press/v80/hartford18a/hartford18a.pdf); [Maron et al., 2018](https://arxiv.org/pdf/1812.09902.pdf); [Maron et al., 2019](https://arxiv.org/pdf/1905.11136.pdf); [Keriven et al., 2019](https://proceedings.neurips.cc/paper/2019/file/ea9268cb43f55d1d12380fb6ea5bf572-Paper.pdf)), and we believe addressing this challenge would be an interesting future research direction.

---

> > ### Comment · Reviewer_385N · 2021-08-20
> > **Response**
> >
> > I really appreciate the effort, especially in answering my and other reviewers' concerns. I will raise my score to 8. I wish the authors best of luck.

---

### Decision · Program_Chairs · 2021-09-27

**Decision:**

Accept (Poster)

**Comment:**

The paper develops transformers for order permutation invariant data such as sets, graphs  and hypergaphs. Overall, the reviewers and I have enjoyed reading the paper since moving transformers towards structured data is an important research direction. The experimental results presented in the rolling discussion on sharing queries and keys across all equivalence classes as well as the ones comparing with Hyper-SAGNN and S2G have to be included in the final version. They really showed the pros of the proposed approach. While making those changes the authors may also wish to add the clarifications posted in the rolling discussion.